# Clinical characteristics and efficacy of short-course antibiotic therapy for *Staphylococcus aureus* bacteremia in hematological patients

Nuobing Yang,[1,2] Hui Wang,[1,3] Xiaomeng Feng,[1,2] Qingsong Lin,[1,2] Biyun Chen,[4] Yingchang Mi,[1,2] Yizhou Zheng,[1,2] Lugui Qiu,[1,2] Fengkui Zhang,[1,2] Erlie Jiang,[1,2] Mingzhe Han,[1,2] Zhijian Xiao,[1,2] Jianxiang Wang,[1,2] Sizhou Feng[1,2]

**ABSTRACT**   To explore the clinical characteristics and outcomes of *Staphylococcus aureus* bacteremia (SAB) in patients with hematological diseases and evaluate the efficacy of short-course antibiotic therapy for uncomplicated SAB in this group. We performed a retrospective study on hematological adult patients with SAB, including a high proportion of neutropenic patients. Logistic regression models fitted with inverse probability of treatment weighting were employed to evaluate the association between treatment duration and clinical outcomes in patients with uncomplicated SAB. A total of 242 patients infected with SAB were included, of whom 38 (15.7%) were caused by MRSA. The 90-day mortality and 30-day mortality rate were 11.2% ($n = 27$) and 4.5% ($n = 11$), respectively, while the 90-day recurrence rate was 5.4% ($n = 13$). Multivariate analysis indicated that advanced age (odds ratio [OR] = 1.063, $P = 0.004$), relapsed or refractory hematological diseases (OR = 14.439, $P < 0.001$), and polymicrobial infection (OR = 5.102, $P = 0.020$) were independent predictors of 90-day mortality, while MRSA bacteremia was an independent predictor of 30-day mortality (OR = 14.091, $P = 0.009$). Among 191 patients with uncomplicated SAB, 89 patients received short-course (median, 8.0 days; IQR, 7.0-9.0) and 102 received long-course therapy (median, 15.0 days; IQR, 12.0-19.3). In the weighted cohort, the multivariate analysis indicated that a short course of antibiotic treatment showed no significant relation with 90-day mortality (OR = 0.595, $P = 0.486$), 30-day mortality (OR = 0.784, $P = 0.832$) or 90-day recurrence (OR = 1.80, $P = 0.373$). For hematological adult patients infected with SAB, MRSA, and polymicrobial infection associated with poor outcomes. Short-course antibiotic therapy for uncomplicated SAB seemed to yield similar clinical outcomes as long-course one.

**IMPORTANCE**   There is still no consensus on the optimal antibiotic course for *Staphylococcus aureus* bacteremia (SAB) in patients with hematological diseases. Some studies have suggested that short-course antibiotic therapy was feasible for uncomplicated SAB, but few have targeted hematological patients. Here, we described clinical characteristics and outcomes of SAB in hematological patients and highlighted advanced age, refractory or relapsed hematological diseases, and polymicrobial infection as independent predictors of 90-day mortality, while MRSA bacteremia was associated with early mortality risk. And we demonstrated that short-course antibiotic therapy (≤10 days) for uncomplicated SAB in hematological patients was non-inferior to long-course one (>10 days).

**KEYWORDS**   hematological diseases, neutropenia, *Staphylococcus aureus* bacteremia, short-course antibiotic therapy

Address correspondence to Sizhou Feng, szfeng@ihcams.ac.cn, or Biyun Chen, 253934019@qq.com.

Nuobing Yang and Hui Wang contributed equally to this article. The author order was determined by who initiated the project and the length of time each author spent on the project.

The authors declare no conflict of interest.

See the funding table on p. 13.

There has been a hot debate about the optimal course of antibiotic therapy in patients with *Staphylococcus aureus* bacteremia (SAB). Current guidelines recommend at least 14 days of antibiotic treatment, but the available evidence supporting the recommendations is scarce and is usually based on individual experts' perspectives (1, 2). Several observational studies have investigated whether less than 14 days of antibiotic therapy was feasible for patients with uncomplicated SAB (3–6). The results indicated that short courses of antibiotic therapy did not appear to increase the mortality or relapse rate for uncomplicated SAB. However, there has been no similar research targeting hematological patients.

As a specific group, hematological adult patients who are infected with bacteremia are more prone to be exposed to prolonged courses of antibiotics due to their immunocompromised status caused by the underlying hematological diseases, chemotherapy or immunosuppressive therapy, and hematopoietic stem cell transplantation (HSCT). However, prolonged exposure to antibiotics is correlated with increased rates of adverse drug events and may lead to the development of multidrug-resistant organisms (7). Our previous research indicated that short-course therapy (7–11 days) for *Pseudomonas aeruginosa* bacteremia was non-inferior to prolonged-course therapy (12–21 days) in terms of 30-day recurrent infection or mortality ($P = 0.979$), and 90-day recurrent infection ($P = 0.139$)(8). What is more, there have been studies indicating that neutropenia at the onset of SAB did not increase the risk of mortality or metastatic infection in hematological patients with SAB (9, 10). Therefore, we want to explore whether it is possible to shorten the antibiotic course of SAB in hematological adult patients without compromising the curative efficacy.

In this study, we determined the clinical characteristics and outcomes of SAB in adult patients with hematological diseases. Meanwhile, by using inverse probability of treatment weighting (IPTW) and limiting immortal time bias, we evaluated the efficacy of ≤10 days versus >10 days of antibiotic therapy on clinical outcomes in patients with uncomplicated SAB, among whom neutropenic patients accounted for a high proportion.

## MATERIALS AND METHODS

### Setting and patients

This retrospective study was conducted at a blood diseases hospital in Tianjin, China, between January 2013 and December 2023. Patients were eligible if they were ≥14 years old, suffering from hematological diseases, and diagnosed with SAB (Fig. 1). Patients meeting any of the following conditions were further excluded from the group of uncomplicated SAB: (i) inability to complete the planned course of therapy due to death or loss to follow-up; (ii) receiving ≥28 days of antibiotic treatment which may indicate a clinical suspicion of complicated infection; (iii) complicated SAB defined by the presence of any of the following: evidence of metastatic infections (such as endocarditis, osteomyelitis, and spondylodiscitis), septic shock, infection involving an implanted prosthesis, a positive blood culture for *S. aureus* obtained after more than 48 h of effective antibiotic treatment; and (iv) polymicrobial bacteremia. In addition, an included patient who experienced a second SAB within 90 days of the first episode could not be included a second time. In fact, the second episode was considered a recurrence. This study was approved by the ethical committee of the Institute of Hematology and Blood Diseases Hospital, Chinese Academy of Medical Sciences and Peking Union Medical College.

### Exposure and outcomes

Duration of appropriate antibiotic therapy was classified as either ≤10 days (short course) or >10 days (long course). Appropriate antibiotics were defined as intravenous antibiotics that the isolated pathogen was susceptible to *in vitro* or oral Contezolid (11, 12). The

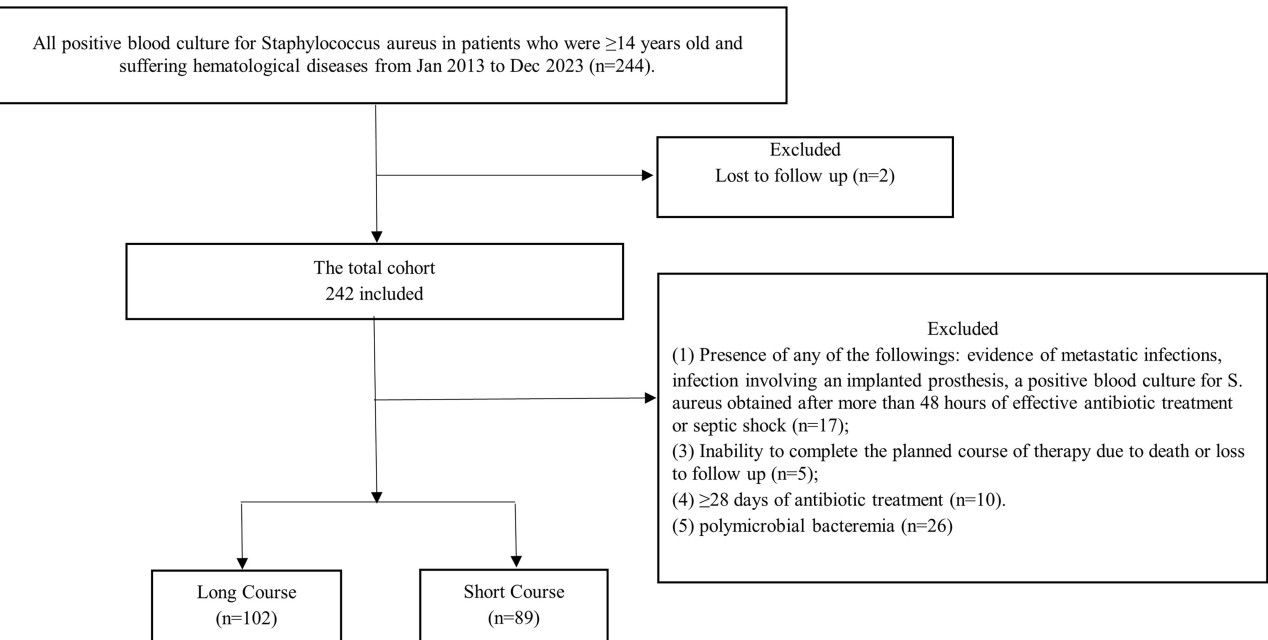

**FIG 1** Study flowchart of *Staphylococcus aureus* bacteremia episodes among hematologic patients receiving short-course versus long-course antibiotic therapy.

primary outcome was all-cause mortality within 90 days of the first positive blood culture. The second outcome included 30-day mortality and 90-day recurrence, which was defined as identification of any microbiologically confirmed *S. aureus* infection within 90 days of the onset of initial SAB, but after stopping antibiotic therapy (4). In our center, indications for discontinuation of antibiotic therapy included: resolution of infection-related signs and symptoms; negative blood cultures; successful control or removal of the identified source of infection (e.g., abscess drainage or excision, removal of the peripherally inserted central catheter [PICC]); normalization of inflammatory markers and neutropenia recovery (absolute neutrophil count [ANC] > 0.5 × 10^9 cells/L). A few patients with persistent neutropenia but improving infection manifestations and controlled infection sources might discontinue antibiotics if their neutrophils were rising, even if not above 0.5 × 10^9 cells/L, as per the clinicians' decisions.

## Clinical data collection

Information regarding underlying hematological disease and its phase, management (chemotherapy, immunosuppressive therapy, and HSCT), diabetes status, Charlson comorbidity index (CCI), ANC at the onset of SAB and at the day of discounting antibiotics (if not available, the ANC from the previous or subsequent day can be taken as an alternative), duration of neutropenia before and after SAB, sources of infection, site of acquisition, complications, septic shock, microbiology data (methicillin-susceptible/resistant), results of follow-up blood cultures (recommended to be performed 48–96 h after antibiotic initiation [13]), antibiotic treatment regimens, and clinical outcomes were collected by chart review for all patients.

Acquisition of SAB was stratified by three categories: (i) community-onset SAB if the positive blood culture was obtained at the time of hospital admission or within 48 h of hospital admission; (ii) nosocomial SAB if the positive blood culture was obtained from patients who had been hospitalized for 48 h or longer; and (iii) healthcare-associated SAB if the patient had regular hospital visits, received specialized nursing care, or was a resident at a nursing home. The history of HSCT within 1 year prior to this episode of SAB or presence of acute or chronic graft-versus-host diseases (GVHD) during the SAB was included in the statistics. Neutropenia was defined as ANC < 0.5 × 10^9 /L. Septic shock was defined as having a systolic pressure <90 mmHg despite adequate fluid resuscitation

or requiring vasopressor agents, which was assessed and managed by the treating physicians based on institutional protocols and international sepsis guidelines (14). The clinical source of infection was defined as the site that was most likely responsible for seeding of *S. aureus* into the bloodstream based on clinical signs, microbiological findings, and imaging results (15), which was initially determined by the treating physicians during clinical management and recorded in the medical charts, and further confirmed by the research team. In cases of uncertainty, adjudication was performed by two independent investigators. For patients with an eradicable focus of SAB, appropriate source control measures, such as abscess drainage or excision and catheter removal, were implemented. Metastatic infection was defined as definite infective endocarditis fulfilling the modified Duke criteria (16) or the development of secondary infection at a site distant from the initial focus of infection (17). Empirical treatment with agents active against aerobic gram-positive cocci was defined as administration of agents including vancomycin, linezolid, teicoplanin, and daptomycin before the susceptibility results were obtained (18).

## Microbiological studies

Clinical samples were processed in the microbiology laboratory of the hospital using an automated system (VITEK 2 Compact) for strain identification and routine drug sensitivity test. Antibiotic susceptibilities were defined according to current Clinical and Laboratory Standards Institute criteria.

## Statistical analyses

Categorical variables were compared using chi-square test or Fisher's exact test, as appropriate, while continuous variables were assessed by the Wilcoxon rank-sum test. Patients who died before having the chance of completing the whole course of therapy were excluded to account for immortal time bias.

Inverse probability of treatment weighting (IPTW) was performed to balance baseline characteristics between short-course and long-course antibiotic groups (19, 20). A pseudo-population was created through assigning individuals with weights that corresponded to the inverse of their probability of receiving short or long treatment given observed covariates. Covariates used for generating propensity scores included time of the index blood culture, age, gender, diabetes, CCI, hematological diseases and its stage, history of HSCT, chemotherapy or immunosuppressive therapy, antibiotic use prior to SAB, *methicillin-resistant S. aureus* (MRSA), source of infection, nosocomial infection, complications (including oral mucositis, perianal infection, and pneumonia), empirical treatment with agents active against aerobic gram-positive cocci, neutropenia, duration of neutropenia before SAB. The covariates were chosen based on previous studies showing an effect on the clinical outcome and could be acquired at the onset of SAB (9, 21, 22). The differences of covariates between the two groups were considered insignificant if the standardized mean difference (SMD) values were <0.1. Finally, odds ratios (ORs) and 95% confidence intervals (CIs) for the outcomes were estimated using weighted logistics regression. $P$ value < 0.05 was considered statistically significant. Data analyzes were performed using R software version 4.3 and SPSS version 26.0.

## RESULTS

Over 11 years, 242 adult patients (≥14 years) with hematological diseases infected with SAB were included in our study. Baseline clinical characteristics were shown in Table 1. Patients had a median age of 34.5 years and 157 (64.9%) were male. The most common hematologic disease was acute myeloid leukemia (AML) ($n$ = 107, 44.2%), followed by acute lymphoblastic leukemia (ALL) ($n$ = 61, 25.2%), aplastic anemia ($n$ = 23, 9.5%), lymphoma ($n$ = 16, 6.6%), and myelodysplastic syndromes ($n$ = 15, 6.2%). Over 90% ($n$ = 219) of the patients underwent chemotherapy or immunosuppressive therapy according to their underlying diseases within 1 month prior to the onset of SAB. Nineteen patients

**TABLE 1** Characteristics of the total population with SAB[a]

| | Death within 90 days of the onset SAB | | |
|---|---|---|---|
| | Yes (n = 27) | No (n = 215) | P |
| Time | | | 0.227 |
| 2012–2017 (%) | 9 (33.3) | 98 (45.6) | |
| 2018-2023 (%) | 18 (66.7) | 117 (54.4) | |
| Age (years), median (IQR) | 46.0 [37.0, 65.0] | 33.0 [21.0, 46.0] | <0.001 |
| Male sex | 17 (63.0) | 140 (65.1) | 0.825 |
| Diabetes mellitus | 3 (11.1) | 16 (7.4) | 0.773 |
| CCI (median [IQR]) | 2.0 [2.0, 2.0] | 2.0 [2.0, 2.0] | 0.067 |
| Type of hematologic disease (%) | | | 0.032 |
| Acute myeloid leukemia | 7 (25.9) | 100 (46.5) | |
| Acute lymphoblastic leukemia | 7 (25.9) | 54 (25.1) | |
| Lymphoma | 5 (18.5) | 11 (5.1) | |
| Others | 8 (29.6) | 50 (23.3) | |
| Stage of underlying diseases (%) | | | <0.001 |
| Induction | 8 (29.6) | 80 (37.2) | |
| Consolidation | 3 (11.1) | 121 (56.3) | |
| Relapsed/refractory | 16 (59.3) | 14 (6.5) | |
| History of HSCT | | | |
| Auto-HSCT (%) | 0 (0.0) | 4 (1.9) | 1.000 |
| Allo-HSCT (%) | 3 (11.1) | 16 (7.4) | 0.773 |
| Chemotherapy or immunosuppressive therapy within 1 month prior to SAB | 19 (70.4) | 200 (93.0) | 0.001 |
| Antibiotic use within 2 months of SAB | 12 (44.4) | 102 (47.4) | 0.769 |
| MRSA (%) | 7 (25.9) | 31 (14.4) | 0.205 |
| Site of infection | | | 0.776 |
| Primary/unknown | 14 (51.9) | 127 (59.1) | |
| Skin/soft tissue | 8 (29.6) | 55 (25.6) | |
| Catheter-associated | 1 (3.7) | 11 (5.1) | |
| Others | 4 (14.8) | 22 (10.2) | |
| Nosocomial infection (%) | 21 (77.8) | 196 (91.2) | 0.069 |
| Complications (%) | 15 (55.6) | 106 (49.3) | 0.540 |
| Empirical treatment with agents active against aerobic gram-positive cocci (%) | 7 (25.9) | 26 (12.1) | 0.094 |
| Polymicrobial bacteremia | 8 (29.6) | 18 (8.4) | 0.002 |
| Metastatic infection | 1 (3.7) | 12 (5.6) | 1.000 |
| Day 1 ANC 0–500 cells/mL | 17 (63.0) | 164 (76.3) | 0.133 |
| Duration of neutropenia before BSI (median [IQR]) | 3.0 [0.0, 5.0] | 3.0 [1.0, 6.0] | 0.566 |
| Duration of neutropenia after BSI (median [IQR]) | 7.0 [0.0, 21.0] | 5.0 [2.0, 11.0] | 0.323 |
| ANC 0–500 cells/mL at the day of discontinuation of antibiotics | 10 (37.0) | 38 (17.7) | 0.017 |
| Complicated SAB | 4 (14.8) | 23 (10.7) | 0.752 |
| Duration of antibiotic treatment (median [IQR]) | 15.0 [7.0, 24.0] | 11.0 [9.0, 17.0] | 0.348 |
| Short course of antibiotic treatment | 10 (37.0) | 93 (43.3) | 0.538 |

[a]allo-HSCT, allogeneic hematopoietic stem cell transplantation; ANC, absolute neutrophil count; auto-HSCT, autologous hematopoietic stem cell transplantation; CCI, Charlson index; HSCT, hematopoietic stem cell transplantation; MRSA, methicillin-resistant *Staphylococcus aureus*; SAB, *Staphylococcus aureus* bacteremia.

had undergone allogeneic HSCT (allo-HSCT) (11 of whom presented with GVHD when the SAB occurred), and 4 patients underwent autologous HSCT (auto-HSCT). At the time of SAB onset, 181 (74.8%) patients had neutropenia, including 154 (63.6%) with severe neutropenia. All patients received bedside consultations from infectious diseases physicians. And 37 (15.3%) patients underwent echocardiography.

Most patients (n = 217, 89.7%) acquired SAB from hospital, and MRSA bacteremia was identified in 38 patients (15.7%). Leading sources of infection were primary (n = 141,

58.3%) and soft tissue or skin ($n$ = 63, 26.0%), followed by respiratory tract-associated ($n$ = 15, 6.2%), PICC-associated ($n$ = 12, 5.0%), and gastrointestinal tract-associated infection ($n$ = 9, 3.7%). In addition, one patient had a splenic abscess identified as the source of SAB and subsequently underwent splenectomy. SAB was considered complicated in 27 patients (11.2%). Metastatic infection occurred in 13 (5.4%) patients, among which skin and soft tissue was the most common site ($n$ = 9, 69.2%) of metastatic infection confirmed by ultrasonography and microbiological evidence, followed by lung ($n$ = 6, 46.2%) confirmed by chest CT and/or bronchoalveolar lavage culture. Two of the patients had the infection disseminated to both the skin and lungs. Of note, there was no case of metastatic endocarditis, osteomyelitis, or other deep-seated *S. aureus* infections in our cohort.

Only 33 patients were empirically administered vancomycin or other agents active against aerobic gram-positive cocci. Three patients developed septic shock during SAB, of whom 1 patient died of septic shock on the third day of SAB. The remaining two patients' shock status was corrected within two days using vasopressor agents and antibiotics, and their courses of antibiotics were 27 and 38 days, respectively. The sensitive antibiotics administered are shown in Table S1. The 90-day mortality and 30-day mortality rates were 11.2% ($n$ = 27) and 4.5% ($n$ = 11), respectively, while the 90-day recurrence rate was 5.4% ($n$ = 13). Multivariate analysis indicated that advanced age (OR = 1.063, $P$ = 0.004), relapsed or refractory hematological diseases (OR = 14.439, $P$ < 0.001), and polymicrobial infection (OR = 5.102, $P$ = 0.020) were independent predictors of 90-day mortality, while MRSA bacteremia was an independent predictor of 30-day mortality (OR = 14.091, $P$ = 0.009). The neutropenic state at the onset of SAB did not contribute to the unfavorable outcome. Univariate and multivariate analyses on the clinical outcomes of the total population were displayed in Table 2. Clinical characteristics between neutropenic and non-neutropenic patients were also compared (Table S2). The incidence of MRSA bacteremia (17.1% [$n$ = 31] vs 11.5% [$n$ = 7], $P$ = 0.294) and complicated SAB (9.4% [$n$ = 17] vs 16.4% [$n$ = 10], $P$ = 0.133) was comparable between neutropenic and non-neutropenic groups. But neutropenic patients were younger, more likely to have a history of chemotherapy or immunosuppressive therapy and antibiotic application prior to this episode of SAB, had more AML, and more nosocomial and primary infections.

After excluding non-eligible episodes, 191 patients met eligibility criteria and were included in the uncomplicated SAB group, of whom 89 (46.6%) received short-course antibiotic therapy (median, 8.0 days; IQR, 7.0–9.0) and 102 (53.4%) received long-course therapy (median, 15.0 days; IQR, 12.0–19.3). Clinical characteristics of the unweighted and weighted cohorts are shown in Table 3. Baseline clinical characteristics of the weighted cohort were well balanced (SMD <0.1). In the short-course group, five (5.6%) died within 90 days after the onset of SAB, while in the long-course group, nine (8.8%) patients did ($P$ = 0.396). Meanwhile, 30-day death occurred in three (3.4%) patients in the short-course group and in three (2.9%) patients in the long-course one ($P$ = 1.000), and 90-day recurrence occurred in eight (9.0%) patients in the short-course group and in four (3.9%) patients in the long-course group ($P$ = 0.254). In the weighted cohort, the univariate and multivariate analyses indicated that a short course of antibiotic treatment was not significantly associated with poor clinical outcomes (Table 4; Table S3). The subgroup analysis of survival is displayed in Fig. 2, demonstrating the non-inferiority of short-course antibiotic therapy compared with long-course ones in multiple subgroups.

## DISCUSSION

Our study showed that neutropenia at the onset of SAB was not significantly associated with poor outcomes in patients with hematological diseases. However, advanced age, relapsed or refractory hematological disease, polymicrobial bacteremia, and MRSA infection were identified as risk factors for mortality in these patients. And in patients with uncomplicated SAB, after adjustment for confounding variables by IPTW and limiting immortal time bias, short-course antibiotic therapy (median, 8 days) conferred

**TABLE 2** Univariate and multivariate analyses on the clinical outcomes of total population[a]

| | Death within 90 days of the onset SAB | | | Death within 30 days of the onset SAB | | | Recurrent SAB infection | | |
|---|---|---|---|---|---|---|---|---|---|
| | Univariate | Multivariate | P | Univariate | Multivariate | P | Univariate | Multivariate | P |
| Age (years), median (IQR) | 1.053 (1.026, 1.082) | 1.063 (1.019, 1.108) | 0.004 | 1.053 (1.013, 1.095) | 1.039 (0.992, 1.089) | 0.106 | 0.986 (0.951, 1.023) | –[b] | – |
| Male sex | 0.911 (0.397, 2.088) | | | 1.468 (0.379, 5.684) | | | 1.231 (0.368, 4.124) | – | – |
| Diabetes mellitus | 1.555 (0.422, 5.726) | | | 1.183 (0.143, 9.772) | | | | – | – |
| CCI (median [IQR]) | 1.911 (0.922, 3.962) | | | 3.650 (1.278, 10.425) | 1.106 (0.370, 3.304) | 0.857 | 0.807 (0.371, 1.757) | – | – |
| Type of hematologic disease (%) | | | | | | | | | |
| Acute myeloid leukemia | 1 | 1 | | 1 | 1 | | 1 | – | – |
| Acute lymphoblastic leukemia | 1.852 (0.617, 5.556) | 4.036 (0.897, 18.152) | 0.069 | 0.873 (0.155, 4.910) | | | 2.299 (0.593, 8.908) | – | – |
| Lymphoma | 6.494 (1.760, 23.961) | 2.357 (0.394, 14.098) | 0.347 | 1.717 (0.180, 16.406) | | | 1.717 (0.180, 16.406) | – | – |
| Others | 2.286 (0.784, 6.662) | 0.551 (0.117, 2.602) | 0.452 | 1.907 (0.459, 7.927) | | | 1.405 (0.303, 6.501) | – | – |
| Stage of underlying diseases (%) | | | | | | | | | |
| Induction | 1 | 1 | | 1 | 1 | | 1 | – | – |
| Consolidation | 0.248 (0.064, 0.963) | 0.467 (0.101, 2.156) | 0.329 | 0.171 (0.019, 1.554) | 0.334 (0.030, 3.672) | 0.370 | 0.420 (0.133, 1.331) | – | – |
| Relapsed/refractory | 11.429 (4.116, 31.729) | 14.439 (3.931, 53.039) | <0.001 | 5.250 (1.369, 20.131) | 4.042 (0.803, 20.351) | 0.090 | – | – | – |
| History of HSCT | 1.219 (0.337, 4.407) | | | – | | | 1.801 (0.374, 8.674) | – | – |
| Chemotherapy or immunosuppressive therapy within 1 month prior to SAB | 0.178 (0.067, 0.474) | 0.080 (0.013, 0.496) | 0.007 | 0.157 (0.042, 0.584) | 0.265 (0.048, 1.464) | 0.128 | 1.275 (0.158, 10.278) | – | – |
| Antibiotic use within 2 months of SAB | 0.886 (0.396, 1.982) | | | 0.405 (0.105, 1.567) | | | 0.481 (0.144, 1.606) | – | – |
| MRSA (%) | 2.077 (0.811, 5.324) | | | 5.000 (1.443, 17.325) | 14.091 (1.963, 101.173) | 0.009 | 2.549 (0.743, 8.746) | – | – |
| Site of infection | | | | | | | | | |
| Primary/unknown | 1 | | | 1 | 1 | | 1 | – | – |
| Skin/soft tissue | 1.319 (0.523, 3.326) | | | 0.892 (0.168, 4.725) | 0.759 (0.091, 6.330) | 0.799 | 1.525 (0.415, 5.606) | – | – |
| Catheter-associated | 0.825 (0.099, 6.872) | | | – | – | | 2.045 (0.226, 18.542) | – | – |
| Others | 1.649 | | | 4.945 | 8.242 | 0.050 | 1.875 | – | – |

*(Continued on next page)*

TABLE 2 Univariate and multivariate analyses on the clinical outcomes of total population[a] (Continued)

| | Death within 90 days of the onset SAB | | | Death within 30 days of the onset SAB | | | Recurrent SAB infection | | |
|---|---|---|---|---|---|---|---|---|---|
| | Univariate | Multivariate | P | Univariate | Multivariate | P | Univariate | Multivariate | P |
| | (0.497, 5.475) | | | (1.232, 19.851) | (1.002, 67.834) | 0.324 | (0.357, 9.843) | – | – |
| Nosocomial infection (%) | 0.339 (0.122, 0.943) | 1.757 (0.285, 10.838) | 0.544 | 1.159 (0.142, 9.455) | – | – | – | – | – |
| Complications (%) | 1.285 (0.575, 2.874) | | | 0.557 (0.159, 1.954) | | | 0.609 (0.193, 1.917) | – | – |
| Empirical treatment with agents active against aerobic gram-positive cocci (%) | 2.544 (0.981, 6.601) | | | 0.622 (0.077, 5.024) | | | 0.513 (0.064, 4.082) | – | – |
| Polymicrobial bacteremia | 4.608 (1.770, 11.996) | 5.102 (1.298, 20.055) | 0.020 | 0.824 (0.101, 6.710) | | | 0.680 (0.085, 5.453) | – | – |
| Metastatic infection | 0.651 (0.081, 5.210) | | | – | | | – | – | – |
| Day 1 ANC 0–500 cells/mL | 0.529 (0.228, 1.227) | | | 0.573 (0.162, 2.030) | | | 0.369 (0.119, 1.144) | – | – |
| Duration of neutropenia before BSI (median [IQR]) | 1.011 (0.975, 1.048) | | | 1.020 (0.976, 1.067) | | | 0.992 (0.928, 1.061) | – | – |
| Duration of neutropenia after BSI (median [IQR]) | 1.009 (0.988, 1.030) | | | 0.990 (0.934, 1.049) | | | 1.007 (0.980, 1.036) | – | – |
| ANC 0–500 cells/mL at the day of discontinuation of antibiotics | 2.740 (1.164, 6.450) | 1.884 (0.520, 6.823) | 0.335 | 3.643 (1.063, 12.492) | 2.116 (0.477, 9.383) | | 3.816 (1.220, 11.940) | – | – |
| Complicated SAB | 1.452 (0.461, 4.569) | | | 0.788 (0.097, 6.411) | | | 0.651 (0.081, 5.210) | – | – |
| Short course of antibiotic treatment | 0.772 (0.338, 1.763) | | | 3.818 (0.987, 14.763) | | | 2.257 (0.716, 7.112) | – | – |

[a]ANC, absolute neutrophil count; CI, confidence interval; MRSA, methicillin-resistant Staphylococcus aureus; OR, odds ratio; SAB, Staphylococcus aureus bacteremia.
[b]'–' indicates not estimable due to zero events in one group or computational limitations, making valid calculation of odds ratios or confidence intervals not possible.

**TABLE 3** Baseline characteristics of patients with uncomplicated SAB before and after inverse probability of treatment weighting[a]

| Characteristic | Long course (n = 102) | Short course (n = 89) | P (before IPTW) | SMD (before IPTW) | P (after IPTW) | SMD (after IPTW) |
|---|---|---|---|---|---|---|
| Time | | | 0.593 | 0.099 | 0.976 | 0.005 |
| 2012–2017 (%) | 42 (41.2) | 41 (46.1) | | | | |
| 2018-2023 (%) | 60 (58.8) | 48 (53.9) | | | | |
| Age (median [IQR]) | 34.50 [21.00, 51.50] | 33.00 [22.00, 44.00] | 0.677 | 0.089 | 0.738 | 0.021 |
| Male (%) | 68 (66.7) | 57 (64.0) | 0.82 | 0.055 | 0.601 | 0.082 |
| Diabetes mellitus (%) | 11 (10.8) | 3 (3.4) | 0.092 | 0.292 | 0.695 | 0.065 |
| CCI (median [IQR]) | 2.00 [2.00, 2.00] | 2.00 [2.00, 2.00] | 0.91 | 0.044 | 0.598 | 0.09 |
| Type of hematologic disease, n (%) | | | 0.414 | 0.248 | 0.999 | 0.023 |
| Acute myeloid leukemia | 42 (41.2) | 44 (49.4) | | | | |
| Acute lymphoblastic leukemia | 24 (23.5) | 22 (24.7) | | | | |
| Lymphoma | 6 (5.9) | 6 (6.7) | | | | |
| Others | 30 (29.4) | 17 (19.1) | | | | |
| Stage of underlying diseases (%) | | | 0.706 | 0.122 | 0.895 | 0.071 |
| Induction | 37 (36.3) | 33 (37.1) | | | | |
| Consolidation | 52 (51.0) | 48 (53.9) | | | | |
| Relapsed/refractory | 13 (12.7) | 8 (9.0) | | | | |
| HSCT (%) | 10 (9.8) | 2 (2.2) | 0.065 | 0.322 | 0.654 | 0.08 |
| Chemotherapy or immunosuppressive therapy within 1 month prior to SAB (%) | 93 (91.2) | 82 (92.1) | 1 | 0.035 | 0.913 | 0.018 |
| Antibiotic use within 2 months of SAB | 46 (45.1) | 45 (50.6) | 0.543 | 0.11 | 0.99 | 0.002 |
| MRSA (%) | 15 (14.7) | 13 (14.6) | 1 | 0.003 | 0.766 | 0.049 |
| Sites of infection (%) | | | 0.951 | 0.086 | 0.976 | 0.069 |
| Primary/unknown | 60 (58.8) | 54 (60.7) | | | | |
| Skin/soft tissue | 24 (23.5) | 22 (24.7) | | | | |
| Catheter-associated | 6 (5.9) | 4 (4.5) | | | | |
| Others | 12 (11.8) | 9 (10.1) | | | | |
| Nosocomial infection (%) | 92 (90.2) | 81 (91.0) | 1 | 0.028 | 0.943 | 0.011 |
| Complications (%) | 53 (52.0) | 39 (43.8) | 0.328 | 0.163 | 0.785 | 0.042 |
| Empirical treatment with agents active against aerobic gram-positive cocci (%) | 17 (16.7) | 6 (6.7) | 0.06 | 0.312 | 0.836 | 0.035 |
| Day 1 ANC 0–500 cells/mL (%) | 77 (75.5) | 67 (75.3) | 1 | 0.005 | 0.633 | 0.078 |
| Duration of neutropenia before SAB (median [IQR]) | 3.00 [0.00, 6.00] | 3.00 [1.00, 6.00] | 0.672 | 0.127 | 0.851 | 0.042 |

[a]ANC, absolute neutrophil count; CCI, Charlson index; HSCT, hematopoietic stem cell transplantation; MRSA, methicillin-resistant *Staphylococcus aureus*; SAB, *Staphylococcus aureus* bacteremia.

similar risks of 90-day death, 30-day death, and 90-day SAB recurrence compared with long-course therapy (median, 15 days).

In our cohort, MRSA bacteremia accounted for 15.7%, lower than the general population with SAB, where the proportion of MRSA bacteremia is 20–50% (23–25). The incidence of metastatic infection (5.4%) was also lower than in the general population. Additionally, the major sites of metastasis in our patients were skin and soft tissues and lungs, with no cases of metastatic endocarditis or osteomyelitis observed. In a study of 293 patients from the general population with SAB, 45 (15.4%) developed metastatic infections and 10 (3.4%) had endocarditis (26). Among them, 82 (28.0%) died during hospitalization, with 68 (83%) of these deaths occurring within the first 30 days after

**TABLE 4** Multivariate logistic regression analysis on the clinical outcomes of the weighted cohort[a]

| | Death within 90 days of the onset SAB | | Death within 30 days of the onset SAB | | Recurrent SAB infection | |
|---|---|---|---|---|---|---|
| | OR (95% CI) | P | OR (95% CI) | P | OR (95% CI) | P |
| Short course of antibiotic therapy | 0.595 (0.137, 2.58) | 0.486 | 0.784 (0.082, 7.53) | 0.832 | 1.80 (0.49, 6.63) | 0.373 |

[a]OR, odds ratio; CI, confidence interval.

the onset of SAB (26). Russell et al. also reported findings from 464 SAB patients with various underlying conditions, not limited to hematologic diseases (27). They identified 134 (28.9%) participants with apparent metastatic foci, with vertebral osteomyelitis (*n* = 54, 11.6%) being the most common, followed by endocarditis (*n* = 35, 7.5%) and septic arthritis (*n* = 29, 6.3%) (27). In their study, the recurrence rate and 90-day all-cause mortality rate were 1.9% (*n* = 9) and 28.0% (*n* = 130), respectively (27). The incidence of adverse clinical outcomes in both of these studies was higher than in our cohort, where the 90-day mortality rate was 11.2% (*n* = 27), 30-day mortality rate was 4.5% (*n* = 11), and the 90-day recurrence rate was 5.4% (*n* = 13). These results indicated that the clinical outcomes of SAB in hematological patients were not inferior to those in non-hematological patients.

Several researchers have studied the impact of neutropenia at the onset of SAB on the clinical outcomes of patients. Ryu et al. conducted a study including 172 patients with hematological malignancies infected with SAB, among whom 64 were neutropenic and 108 were non-neutropenic at the time of SAB infection detection (9). They found

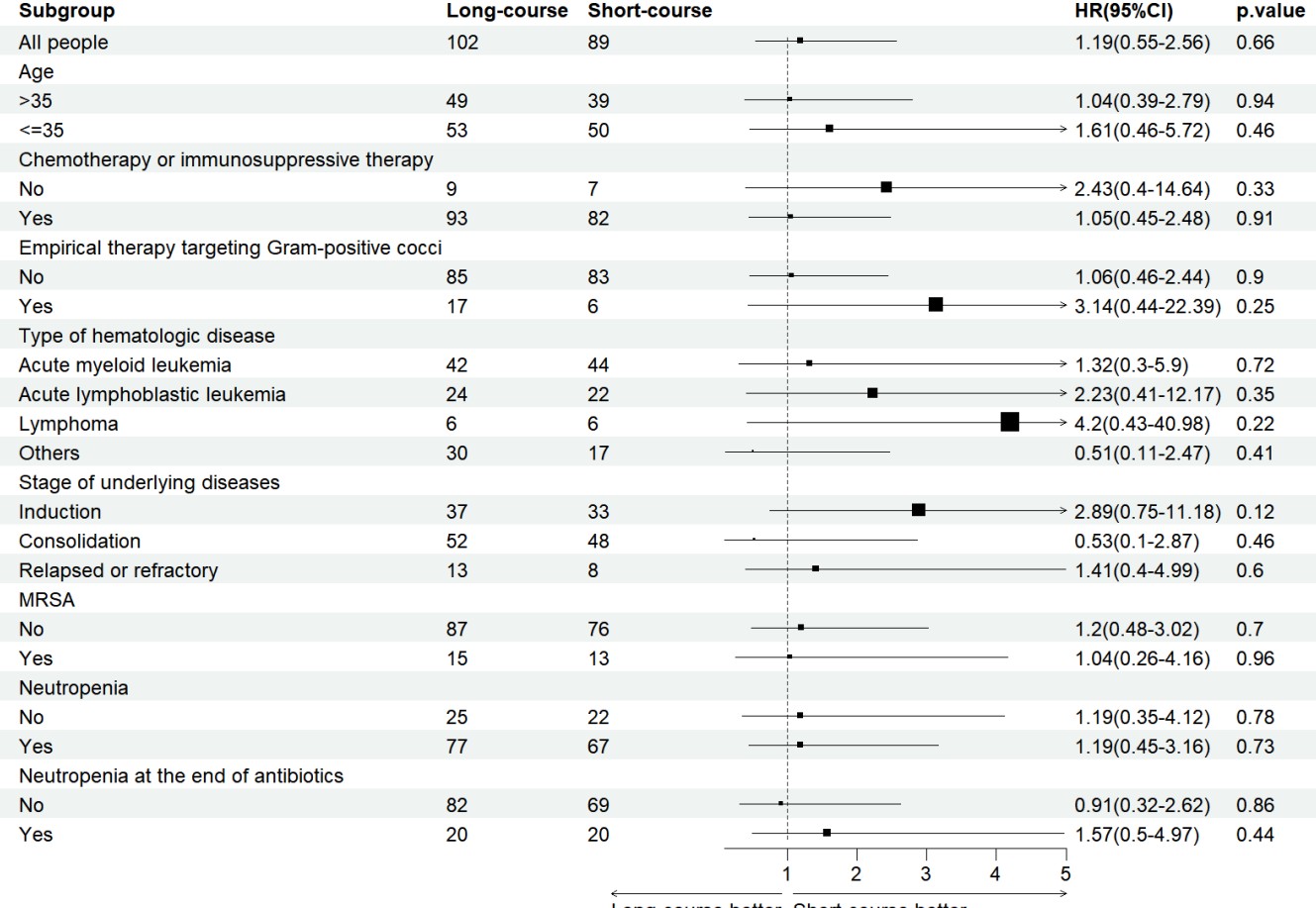

**FIG 2** Subgroup analysis of 90-day poor outcomes (including 90-day death and recurrence) in hematologic patients with *Staphylococcus aureus* bacteremia receiving short-course versus long-course antibiotic therapy.

no significant differences in the incidence of metastatic infection between the two groups (17.2% [11/64] vs 17.6% [19/108], $P = 0.95$) and 12-week mortality in neutropenic patients tended to be lower than in non-neutropenic patients (15.6% [10/64] vs 26.9% [29/108], $P = 0.09$), despite not being statistically significant (9). In the weighted cohort of Camp et al., 83 patients had neutropenia at the time of SAB diagnosis, while 220 patients did not have neutropenia (28). The study showed that SAB in neutropenic patients less frequently resulted in deep-seated metastatic foci such as endocarditis, osteomyelitis, and spondylodiscitis compared to non-neutropenic patients (28). Similarly, in our cohort with a high proportion of neutropenia (74.8%), there were no metastatic cases of endocarditis, osteomyelitis, or other deep-seated metastatic infections. In addition, consistent with the guidelines, empirical treatment with agents active against aerobic gram-positive cocci did not improve clinical outcomes of patients in our study, indicating that vancomycin (or other agents active against aerobic gram-positive cocci) is not necessary as a standard part of initial antibiotic regimen for febrile patients with neutropenia (18).

Given the low incidence of deep-seated metastatic infection, short-term treatment can be feasible for hematologic adult patients with SAB. Guidelines recommended at least 14 days of intravenous antibiotics for uncomplicated SAB, but only based on limited evidence (1). There have been studies investigating the feasibility of short-course antibiotic treatment for patients with SAB. Thorlacius-Ussing et al. compared the effectiveness of short-course (6–10 days) and prolonged-course (11–16 days) treatments in three retrospective observational cohorts of 645, 219, and 141 patients (total 1,005) with low-risk uncomplicated methicillin-susceptible SAB (5). They found no significant differences in the pooled or the independent cohorts in terms of the 90-day mortality (cohort I: OR = 0.85, 95% CI = 0.49–1.41; cohort II: OR = 1.24, 95% CI = 0.60–2.62; cohort III: OR = 1.15, 95% CI = 0.24–4.01; pooled cohort: OR = 1.05, 95% CI = 0.71–1.51), nor the risk of relapse in cohort I (OR = 0.88, 95% CI = 0.54–1.81). Kim et al. evaluated the efficacy of <14 days of antibiotic therapy in 785 patients with uncomplicated SAB and demonstrated its non-inferiority to >14 days of therapy (aOR = 1.24, $P = 0.471$) (6). These studies only included a few hematological patients. A retrospective study investigated the efficacy of short-course antibiotic therapy (<28 days) on uncomplicated SAB targeting immunodeficiency patients but included only 51 patients (29). The sample size was so small, and a 28-day course of antibiotic therapy was not shorter than the recommended duration in current guidelines, thus having little clinical significance. Patients with hematological diseases are more prone to be administered with prolonged courses of antibiotic treatment due to their immunosuppressive status, thus with higher risks of antibiotic-related adverse events and antimicrobial resistance (30, 31). In spite of the great significance of safely decreasing antibiotic exposure for hematological patients, the lack of research data makes it difficult to interpret whether short-course antibiotic therapy is safe for SAB in the group. Our study addresses this gap by targeting adult hematological patients among whom 76.7% had neutropenia. We included patients who received systematic treatment in our center and had accurate follow-up information on mortality and relapse after the onset of SAB, enabling us to make exact assessments of the effect of antibiotic duration on clinical outcomes. In addition, the covariates included in the weighted model were acquired at the onset of SAB based on previous studies showing an effect on the clinical outcome. These made our study more comparable to RCTs and ensured the reliability of the results.

Different from other SAB studies, patients in our cohort were predominantly young, having less comorbidities (in accordance with bacteremia in hematological patients [32]), making our study more applicable to patients in the hematology ward. Regarding the source of SAB, over 50% of the patients in our cohorts had primary SAB with an unidentified source, a phenomenon more evident in neutropenic patients but unrelated to poor outcomes. While prior studies have linked unknown sources to complicated infections (33), this may not hold true for hematologic patients, who typically exhibit a diminished inflammatory response (28).

The study has several strengths. One is adjustment for confounding factors using a propensity score analysis, and another is excluding patients who died or were lost to follow-up before the completion of the anticipated antibiotic treatment course, which limited immortal time bias. Moreover, the integrity of the data and no loss to follow-up enhance the persuasiveness of the study. These criteria approximated our study to the real-world scenario. There are also several limitations that should be acknowledged, mainly due to its retrospective design. First, prolonged courses of antibiotic therapy were more prone to be selectively assigned to patients with more severe infections, which caused the potential bias of confounding by indication. Although we attempted to address the issue by IPTW, which has been reported to could effectively deal with imbalances between research groups with regards to baseline characteristics (34, 35), residual bias may still occur. Also, our study cohort consisted of relatively young patients, predominantly with hospital-acquired infections and fewer underlying diseases (e.g., diabetes, valvular heart disease, or prosthetic implants). Thus, caution is needed when generalizing these findings to all hematologic patients. Finally, not every patient had follow-up blood cultures performed within 48–96 h after initiating antibiotic therapy, which is a typical drawback of retrospective studies. In addition, therapeutic drug monitoring was not systematically implemented during the study period (2013–2023), which may have affected the evaluation of antibiotic efficacy and toxicity. Finally, infection-related mortality was not specifically assessed, and our outcome analysis was limited to all-cause mortality. Further randomized controlled trials are needed to validate our results.

In conclusion, in a population of hematological patients with SAB who have a relatively low rate of MRSA, advanced age, relapsed or refractory hematological diseases, polymicrobial bacteremia, and MRSA infection were independent risk factors for poor clinical outcome. And short-course antibiotic therapy was non-inferior to long-course therapy in treating uncomplicated SAB in hematological patients. The results of the study warrant a randomized controlled trial to further confirm the efficacy and safety of shortened antimicrobial therapy in hematological patients with uncomplicated SAB.

## ACKNOWLEDGMENTS

The authors would like to thank all the reviewers who participated in the review, as well as for providing English editing services during the preparation of this manuscript.

This work was supported by the Chinese Academy of Medical Sciences Innovation Fund for Medical Sciences (2023-I2M-2-007, 2021-I2M-1-017), the National Natural Sciences Foundation of China (82470208), and the National Key R&D Program of China (2024YFC2510500).

N.Y. and H.W. were responsible for the data collection, data curation methodology, and writing the original draft. X.F. contributed to writing and editing. Q.L. contributed to the data collection. E J., Y.M., Y.Z., L.Q., F.Z., M.H., Z.X., and J.W. finished the formal analysis and supervision. B.C. and S.F. supplied the conceptualization, funding acquisition, resources, supervision, and writing—review and editing. All authors reviewed the manuscript.

## AUTHOR AFFILIATIONS

[1]State Key Laboratory of Experimental Hematology, National Clinical Research Center for Blood Diseases, Haihe Laboratory of Cell Ecosystem, Institute of Hematology & Blood Diseases Hospital, Chinese Academy of Medical Sciences & Peking Union Medical College, Tianjin, China

[2]Tianjin Institutes of Health Science, Tianjin, China

[3]Department of Hematology, The Affiliated Yantai Yuhuangding Hospital of Qingdao University, Yantai, China

[4]Department of Hematology, Shengli Clinical Medical College of Fujian Medical University, Fujian Provincial Hospital, Fuzhou, China

## AUTHOR ORCIDs

Biyun Chen  http://orcid.org/0000-0003-3777-7609
Sizhou Feng  http://orcid.org/0000-0003-2768-6690

## FUNDING

| Funder | Grant(s) | Author(s) |
| --- | --- | --- |
| Chinese Academy of Medical Sciences Innovation Fund for Medical Sciences | 2023-I2M-2-007,2021-I2M-1-017 | Sizhou Feng |
| National Natural Sciences Foundation of China | 82470208 | Sizhou Feng |
| National Key R&D Program of China | 2024YFC2510500 | Sizhou Feng |

## AUTHOR CONTRIBUTIONS

Nuobing Yang, Data curation, Formal analysis, Investigation, Methodology, Project administration, Writing – original draft, Writing – review and editing | Hui Wang, Data curation, Writing – original draft, Writing – review and editing | Xiaomeng Feng, Writing – review and editing | Qingsong Lin, Resources | Biyun Chen, Conceptualization, Funding acquisition, Resources, Supervision, Writing – review and editing | Yingchang Mi, Formal analysis, Supervision | Yizhou Zheng, Formal analysis, Supervision | Lugui Qiu, Formal analysis, Supervision | Fengkui Zhang, Formal analysis, Supervision | Erlie Jiang, Formal analysis, Supervision | Mingzhe Han, Formal analysis, Supervision | Zhijian Xiao, Formal analysis, Supervision | Jianxiang Wang, Formal analysis, Supervision | Sizhou Feng, Conceptualization, Funding acquisition, Resources, Supervision, Writing – review and editing

## ETHICS APPROVAL

This study was approved by the Ethics Committee of the Blood Diseases Hospital, Chinese Academy of Medical Sciences. Lot number: IIT2022071-EC-1. Please refer to the attached file. We confirmed that all methods were performed in accordance with the relevant guidelines and regulations.

## ADDITIONAL FILES

The following material is available online.

### Supplemental Material

**Supplemental material (Spectrum02325-24-s0001.docx).** Tables S1 to S3.

### Open Peer Review

**PEER REVIEW HISTORY (review-history.pdf).** An accounting of the reviewer comments and feedback.

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
