## [Reviewer comments · Microbiology Spectrum]

Microbiology Spectrum

Clinical Characteristics and Efficacy of Short-Course Antibiotic Therapy for *Staphylococcus aureus* Bacteremia in Hematological Patients

Nuobing Yang, Hui Wang, Xiaomeng Feng, Qingsong Lin, Biyun Chen, Yingchang Mi, Yizhou Zheng, Lugui Qiu, Fengkui Zhang, Erle Jiang, Mingzhe Han, Zhijian Xiao, Jianxiang Wang, and Sizhou Feng

Corresponding Author(s): Sizhou Feng, Institute of Hematology & Blood Diseases Hospital, Chinese Academy of Medical Sciences & Peking Union Medical College

Review Timeline:

Submission Date:	September 18, 2024
Editorial Decision:	January 27, 2025
Revision Received:	April 5, 2025
Editorial Decision:	May 5, 2025
Revision Received:	May 31, 2025
Accepted:	June 29, 2025

Editor: Bonnie Prokesch

Reviewer(s): The reviewers have opted to remain anonymous.

Transaction Report:

DOI: <https://doi.org/10.1128/spectrum.02325-24>

Re: Spectrum02325-24 (Clinical Characteristics of Staphylococcus Aureus Bacteremia in Adult Hematological Patients and Its Proper Course of Antibiotics)

Dear Prof. Sizhou Feng:

Thank you for the privilege of reviewing your work. Below you will find my comments, instructions from the Spectrum editorial office, and the reviewer comments.

Revision Guidelines

Sincerely,
Bonnie Prokesch
Editor
Microbiology Spectrum

Reviewer #2 (Comments for the Author):

The manuscript titled Clinical Characteristics of Staphylococcus aureus Bacteremia in Adult Hematological Patients and Its Proper Course of Antibiotics by Yang N et al. retrospectively assessed outcomes of patients based on a short versus long course of antibiotics for S. aureus bacteremia. The manuscript is aimed at addressing whether the patient's hematological disease status should be used to exclude the patient from receiving a short course (≤ 10 days) when other factors align with the

infection being uncomplicated.

Major:

1)The currently recommended treatment for *S. aureus* bloodstream infection is at least 14 days. Please comment as to why 10 days was selected for evaluation.

2)Minor editing is recommended to improve grammar and syntax. Examples include: title, lines 81, line 169.

3)Were repeat blood cultures obtained to document clearance in all patients? Generally, duration is established from the last positive blood culture, was that done here or was duration of therapy based on the initial positive culture? If those with repeat positive cultures received longer durations of therapy, then it might bias the short-duration group to have less severe infection and better outcomes. Line 293 speaks to follow-up blood cultures as a limitation, but this likely needs discussed in the methods if repeat blood cultures were collected after the initial positive blood culture and if these were used to make treatment decisions.

4)Results and discussion: The authors claim that neutropenia was "protective... against mortality or relapse" but that is not supported by the data or design of this study. Presence of neutropenia at the day SAB was discovered was used for comparison between those with relapse; however, neutropenia is not a static condition. At best, the question as to how neutropenia at the time bacteremia is discovered plays a role in outcomes could be raised in the discussion.

5)It is interesting that only 17 patients (or 7%) were excluded due to complicated infections. This seems low given the patient population being discussed. IDSA guidelines for MRSA define complicated SAB as presence of locally complicated or metastatic infection, presence of certain risk factors (including prosthetic material, persistent fever, persistently positive blood cultures) and skin findings suggestive of systemic infection. Again, this raises the question as to whether some patients who received longer courses of therapy would meet criteria for complicated infection.

6)It is unclear if polymicrobial bacteremia impacted outcomes. Did the authors consider excluding patients with polymicrobial bacteremia given that empirical and definitive treatment for those organisms was not analyzed?

Minor:

1)Line 68: remove "that" to read, "observational studies have investigated whether..."

2)Line 175: type: "shin" should be "skin"

3)Line 180: The authors cannot determine if administering an empiric agent against *S. aureus* affected the clinical outcome. Please remove that statement.

4)Figures (particularly figure 2): Please add a legend explaining the figure.

Reviewer #3 (Comments for the Author):

Summary

This work aims to provide insight on the impact of treatment duration for uncomplicated SAB in patients with hematologic malignancy.

Major comments

- This study does not appear to account for key factors and standards of care that are known to impact SAB outcomes (ID consultation, bundle/checklist-based patient management, echocardiography, source control [e.g., catheter removal], definitive therapy choice [e.g., beta-lactam vs vancomycin for MSSA SAB] and therapeutic drug monitoring [e.g., appropriate vancomycin levels]).

- Please describe why a primary outcome of death at 90 days was chosen, given the potential for confounding and immortal time bias associated with non-SAB-related causes of death in an acutely ill patient population between the completion of SAB treatment and 90 days following initial culture positivity

Minor comments

- Line 93: Why was an age cut-off of ≥ 14 years old utilized versus a standard adult population of ≥ 18 years old?

- Line 126: Who completed the assessment regarding the adequacy of fluid resuscitation, and how was it assessed?

- Line 127: Who assessed the infection source (treating team, or retrospectively by study team)?

- Lines 142-145: References to support this statistical approach and execution should be included

- Line 147: Why was GVHD (skin, GI, other sites) not included as a covariate?

- Line 149: Why were these complications chosen, when others identified by references studies were not (e.g., catheter removal in Bello-Chavolla, et al)?

- Line 97 & 174: In line 97, patients with metastatic infections were noted as excluded from the definition of uncomplicated SAB.

Please clarify how patients with metastatic infection are reported in the results.

- Table 1: Was neutrophil recovery at the time of therapy cessation included in the study, and if not, please discuss why, given its potential association with treatment duration selection.

Re: Spectrum02325-24 “Clinical Characteristics of Staphylococcus Aureus Bacteremia in Adult Hematological Patients and Its Proper Course of Antibiotics”

Dear editor,

Thanks for providing us with this great opportunity to submit a revised version of our manuscript. We appreciate the detailed and constructive comments provided by the reviewers. Following your suggestions, we have made extensive revisions to our previous draft. The reviewer comments are presented in italicized font below, with specific concerns numbered accordingly. Our responses are in regular font, and the modifications or additions have been incorporated into the revised manuscript. The revisions have been uploaded in the Marked-up Manuscript using track changes. Additionally, figures have been redrawn and re-uploaded.

We hope this revised manuscript has addressed your concerns and look forward to hearing from you.

Sincerely,

The Authors

Email: doctor_szhfeng@163.com; yangnuobing@ihcams.ac.cn

Reply to Reviewer 2

Magor:

Comment 1: *“The currently recommended treatment for S. aureus bloodstream infection is at least 14 days. Please comment as to why 10 days was selected for evaluation.”*

Response 1: Thank you for your valuable comments on our manuscript. We selected 10 days for evaluation based on the following reasons. Firstly, current guidelines recommended at least 14 days of antibiotic treatment for S. aureus bloodstream (SAB) infection, but the available evidence supporting the recommendations is scarce. Thorlacius-Ussing et al. compared the 90-day clinical outcomes of patients receiving short-course (6-10 days) or prolonged-course (11-16 days) antibiotic therapy for low-risk SAB in their study, which comprised the largest study population yet described (n= 1005) in the literature including cases of SAB collected over >2 decades(1). They found 6-10 days of antibiotic treatment was

non-inferior to longer course of antibiotics in terms of 90-day mortality (OR 1.05, 95% CI 0.71–1.51) or the risk of relapse (OR, 0.88, 95% CI 0.54–1.81)(1). Including a small number of immunosuppressive patients, this study did not find immunosuppression to be a contraindication for short-course antibiotic treatment; however, the limited sample size and the lack of categorization of immunosuppression causes restrict its conclusions.

Secondly, rare patients in our cohort had predisposing conditions for metastatic infection, such as history of endocarditis, native valve disease, prosthetic valve, cardiac implantable electronic devices or injection drug use. And the majority of patients in our center had hospital-acquired SAB rather than community-acquired infections, with the latter being classified as a risk factor for complicated SAB(2, 3).

Thirdly, studies have shown that in patients with hematological diseases, neutropenia at the time of SAB detection is not a risk factor for complicated bacteremia. Ryu et al. conducted a study including 64 neutropenic and 108 non-neutropenic patients with hematologic malignancies, and found that there were no significant differences in the incidence of metastatic infection [17.2% (11/64) vs 17.6% (19/108), $p=0.95$](4). Twelve-week mortality in neutropenic patients tended to be lower than in non-neutropenic patients [15.6% (10/64) vs. 26.9% (29/108), $p=0.09$], but not statistically significant(4). Camp et al also compared the outcome of SAB in their weighted cohort, which included 83 neutropenic patients and 220 non-neutropenic patients, and they found no significant difference in mortality between the two groups (36.1% in neutropenic vs. 30.6% in nonneutropenic patients; hazard ratio (HR) 1.21; 95% CI, 0.79-1.83)(5). What's more, the study showed that SAB in neutropenic patients less frequently resulted in deep-seated metastatic foci such as endocarditis, osteomyelitis and spondylodiscitis compared to non-neutropenic patients(5). Similar results were also observed in another study of Venditti et al, which compared outcomes of SAB in 36 neutropenic hematological patients and 36 non-neutropenic hematological patients(6). Early complications such as severe sepsis or septic shock ($p=0.002$) and later ones such as endocarditis and metastatic abscesses ($p=0.02$) were more common in non-neutropenic patient group(6).

Finally, hematological adult patients who are infected with bacteremia are more prone to be exposed to prolonged course of antibiotics due to their immunocompromised status. However, it has been demonstrated that each additional day of antibiotic therapy was associated with an increased risk of adverse events and antimicrobial resistance(7). Therefore, reducing antibiotic exposure safely is crucial for hematologic patients.

Based on these considerations, we selected 10 days for evaluation.

Comment 2: *“Minor editing is recommended to improve grammar and syntax. Examples include: title, lines 81, line 169.”*

Response 2: Thank you for your insightful suggestion. The modifications are as follows:

- 1) The title has been revised to: *“Clinical Characteristics and Efficacy of Short-Course Antibiotic Therapy for Staphylococcus aureus Bacteremia in Hematological Patients”* (Line 3-4 in the Marked-up manuscript); and the running title has been revised to: *“SAB in Hematologic Patients: Short vs. Long Therapy”* (Line 6 in the Marked-up manuscript).
- 2) The original text in line 81 has been revised to: *“What’s more, there have been studies indicating that neutropenia at the onset of SAB did not increase the risk of mortality or metastatic infection in hematological patients with SAB”* (Line 92-94 in the Marked-up manuscript).
- 3) The original text in line 169 has been revised to: *“When SAB occurred, 181 (74.8%) patients had neutropenia, including 154 (63.6%) with severe neutropenia.”* (Line 201-202 in the Marked-up manuscript).

Comment 3: *“Were repeat blood cultures obtained to document clearance in all patients? Generally, duration is established from the last positive blood culture, was that done here or was duration of therapy based on the initial positive culture? If those with repeat positive cultures received longer durations of therapy, then it might bias the short-duration group to have less severe infection and better outcomes. Line 293 speaks to follow-up blood cultures as a limitation, but this likely needs discussed in the methods if repeat blood cultures were collected after the initial positive blood culture and if these were used to make treatment*

decisions.”

Response 3: Thank you for your valuable comments on our manuscript. We appreciate the opportunity to address your concerns and we have added the content of when follow-up blood cultures were performed to the Method section (Line 141-142 in the Marked-up manuscript). Additionally, we also outlined the indications for discontinuing antibiotics at our center (Line 125-132 in the Marked-up manuscript).

Follow-up blood cultures were recommended to be performed 48–96 h after antimicrobial therapy was started(8, 9). In our cohort, 155 (64.0%) patients had follow-up blood cultures performed 48–96 h after initiating antimicrobial therapy, while another 45 patients had their blood cultures repeated more than 96 h after antimicrobial therapy initiation. Patients with repeat positive cultures were classified as having complicated SAB and were consequently excluded from the uncomplicated SAB study cohort, ensuring no impact on the results.

For patients who did not undergo repeat blood cultures or had them performed beyond 96 hours, this was primarily due to rapid clinical improvement, including rapid defervescence, resolution of infection-related symptoms and signs, and prompt control or elimination of the infection source. Since repeat blood cultures have limited utility in most cases of documented bacteremia, particularly in patients already receiving antibiotics(10), the clinical significance of follow-up blood culture results in guiding the duration of antibiotic treatment remains limited. The optimal duration of antibiotic therapy still requires a comprehensive clinical assessment, taking into account underlying diseases, comorbidities, resolution of infection-related manifestations, and infection source control.

Comment 4: *“Results and discussion: The authors claim that neutropenia was "protective... against mortality or relapse" but that is not supported by the data or design of this study. Presence of neutropenia at the day SAB was discovered was used for comparison between those with relapse; however, neutropenia is not a static condition. At best, the question as to*

how neutropenia at the time bacteremia is discovered plays a role in outcomes could be raised in the discussion.”

Response 4: Thank you very much for your suggestions. We deeply recognize the issues in the original manuscript: neutropenia is not a static condition, yet both this study and previous literature have only considered the impact of neutropenia at the time of SAB detection on clinical outcomes. Therefore, we have revised the composite outcome of mortality and recurrence into 90-day mortality, 30-day mortality, and 90-day recurrence (Line 121-125 in the Marked-up manuscript). And the impact of neutropenia at the time of SAB detection on clinical outcomes has been raised in the discussion (Line 287-307 in the Marked-up manuscript). Additionally, we have separately analyzed the impact of neutropenia at the time of SAB onset and antibiotic discontinuation on the outcomes of SAB in hematologic patients (Line 138-140, Table1, Table2 in the Marked-up manuscript). Our results indicated that neutropenia at the time of SAB onset was not associated with complicated infection or poor clinical outcomes. Additionally, persistent neutropenia at the time of antibiotic discontinuation was a predictor for poor outcomes only in the univariate analysis but did not reach statistical significance in the multivariate analysis.

For certain immunosuppressed patients infected with SAB, particularly those with hematologic diseases, prolonged and refractory neutropenia is common. However, merely extending the duration of antibiotic treatment has limited benefits in improving patient prognosis. Hematologic patients undergoing chemotherapy or other treatments often experience hepatic, renal, and gastrointestinal impairment, and prolonged antibiotic use can further exacerbate the impairment due to drug toxicity(11). This suggests that clinicians, when managing such patients, should take additional measures beyond merely extending the antibiotic course, such as improved isolation and care measures, as well as granulocyte-stimulating therapy. This issue has also been added to the Discussion section (Line 349-352 in the Marked-up manuscript).

Comment 5: *“It is interesting that only 17 patients (or 7%) were excluded due to complicated*

infections. This seems low given the patient population being discussed. IDSA guidelines for MRSA define complicated SAB as presence of locally complicated or metastatic infection, presence of certain risk factors (including prosthetic material, persistent fever, persistently positive blood cultures) and skin findings suggestive of systemic infection. Again, this raises the question as to whether some patients who received longer courses of therapy would meet criteria for complicated infection.”

Response 5: Thank you for highlighting this important issue. Reasons for the low proportion of patients with complicated SAB in our cohort are roughly as follows :

Firstly, the patients in our cohort were relatively young, had few comorbidities, and very few had risk factors for complicated bacteremia, such as history of endocarditis, native valve disease, prosthetic valve, or foreign implants. And the majority of them had hospital-acquired SAB rather than community-acquired infections, with the latter being classified as a risk factor for complicated SAB(2).

Secondly, previous studies have found that neutropenic patients with hematologic diseases have a lower rate of developing deep-seated metastatic infections such as endocarditis and osteomyelitis when infected with SAB. This might be due to a lower bacterial inoculum and earlier broad-spectrum antibiotic administration(5, 6). Also, there have been studies indicated that neutrophils carrying *S. aureus* could promote the dissemination of infection, despite a definite conclusion that the incidence of metastatic infections is lower in neutropenic patients has not yet been reached(12-15). These might also be the reasons why the proportion of patients with complicated SAB in our cohort is relatively low.

Thirdly, the definition of complicated SAB in IDSA guidelines includes presence of locally complicated or metastatic infection, or the presence of 1 of several risk factors (presence of prosthetic material, persistent fever, persistently positive blood cultures, and skin findings suggestive of systemic infection). However, associations between these risk factors and occurrence of complicated SAB have been poorly validated. Vaart et al investigated the accuracy of categorizing complicated SAB using the IDSA criteria in their study including 490 patients with SAB(16). They found risk factors in the IDSA definition of complicated

SAB had low to moderate predictive value to identify complicated SAB (a positive predictive value of 70.9% (95% CI, 65.5–75.9) and a negative predictive value of 57.5% (95% CI, 49.1–64.8)), thus their use may lead to unnecessary prolonged antibiotic use(16).

Finally, in addition to 17 cases of confirmed complicated SAB, 10 patients were also excluded from the uncomplicated SAB cohort due to the antibiotic course of more than 28 days. These patients were administrated prolonged antibiotics due to persistent clinical symptoms or recurrent fever during treatment. Blood culture results in these patients may be falsely negative because of antibiotic application, but these cases should still be classified as complicated infections. The correction has been made in the Results section (Line 207 in the Marked-up manuscript).

Comment 6: *“It is unclear if polymicrobial bacteremia impacted outcomes. Did the authors consider excluding patients with polymicrobial bacteremia given that empirical and definitive treatment for those organisms was not analyzed?”*

Response 6: Thank you again for your insightful suggestion. We acknowledge that polymicrobial bacteremia could confound the outcomes, as empirical and definitive treatments for co-pathogens were not analyzed. Based on this consideration, we have now revised our analysis and excluded patients with polymicrobial bacteremia. The corresponding changes have been made in the Methods, Results, and Tables to ensure clarity and consistency. We appreciate your suggestion, which has helped improve the robustness of our study.

Minor:

- 1) *Line 68: remove "that" to read, "observational studies have investigated whether..."*
- 2) *Line 175: type: "shin" should be "skin"*
- 3) *Line 180: The authors cannot determine if administering an empiric agent against S. aureus affected the clinical outcome. Please remove that statement.*
- 4) *Figures (particularly figure 2): Please add a legend explaining the figure.*

Response: We appreciate the reviewer's careful review and valuable suggestions. All requested revisions have been incorporated into the manuscript as follows: (1) 'that' has been removed for improved clarity (Line 80 in the Marked-up manuscript); (2) the typographical error 'shin' has been corrected to 'skin' in line 208 in the Marked-up manuscript; (3) the statement "but it did not affect the clinical outcome" has been removed (Line 214 in the Marked-up manuscript); (4) figure legends, particularly for Figure 2, have been added to enhance clarity (Line 578-579, Line 581-582 in the Marked-up manuscript). These revisions are reflected in the Marked-up Manuscript.

Reply to Reviewer 3:

Major comments

Comment 1: *"This study does not appear to account for key factors and standards of care that are known to impact SAB outcomes (ID consultation, bundle/checklist-based patient management, echocardiography, source control [e.g., catheter removal], definitive therapy choice [e.g., beta-lactam vs vancomycin for MSSA SAB] and therapeutic drug monitoring [e.g., appropriate vancomycin levels])."*

Response 1: Thank you for your insightful suggestion. Our center is a specialized hospital for hematological diseases and given that such patients are highly vulnerable to infections due to immunosuppression, chemotherapy, and hematologic stem cell transplantation therapy, each department is staffed with at least one infectious disease specialist to provide optimal management plans for hematological patients with coexisting infections.

In our center, echocardiography is not required for every patient diagnosed with SAB. In young patients with minimal underlying conditions, no preexisting cardiac disease, and rapid resolution of infection-related manifestations without signs of endocarditis, routine echocardiographic screening is often deemed unnecessary. Instead it is recommended for those with specific indications including predisposing condition for endocarditis (such as native valve disease, prosthetic valve, any cardiac implantable electronic devices), clinical signs of endocarditis (such as embolic events, conjunctival hemorrhages, Janeway's lesions

and immunologic phenomena), cardiac dysfunction, persistent fever and recent hematopoietic stem cell transplantation, which is consistent with current recommendations(17-19). In our cohort, 37 patients (15.3%) underwent echocardiography after the detection of SAB, with no abnormal findings. These patients were also evenly distributed between those who survived and those who did not.

For patients with an eradicable source of SAB, clinicians have implemented appropriate source control measures, such as abscess drainage or excision and catheter removal. The content regarding infection source control measures has been added in Method section (Line 158-159 in the Marked-up manuscript) of the manuscript.

Definitive antibiotic regimens were displayed in Supplementary table 1. The majority of patients in our cohort were given an antibiotic protocol consisting of glycopeptides or linezolid along with anti-pseudomonal β -lactam agents. The latter were employed to prevent and treat infections caused by Gram-negative bacteria, as these are more frequently seen and carry a higher risk of death in hematological patients(20-22). In addition, 11 patients in our cohort were treated with oral Contezolid, a novel oxazolidinone that functions as an inhibitor of bacterial protein synthesis, which was approved by the China National Medical Products Administration in 2021. Multiple studies have reported that Contezolid can be an appropriate alternative for patients who are intolerant to linezolid(23-25). Patients received Contezolid were evenly distributed in both short-course and long-course (1.9% (n=2) vs 6.5% (n=9), p=0.173).

Regarding therapeutic drug monitoring (TDM), it was not systematically implemented during the study period (2013-2023), which we acknowledge as a limitation of our study. However, in the past two years, TDM for vancomycin and teicoplanin has been gradually implemented in our center which may improve antibiotic management in future clinical practice.

Comment 2: *“Please describe why a primary outcome of death at 90 days was chosen, given the potential for confounding and immortal time bias associated with non-SAB-related causes*

of death in an acutely ill patient population between the completion of SAB treatment and 90 days following initial culture positivity.”

Response 2: Thank you for your insightful question. We chose 90-day mortality as the primary outcome because it aligns with previous researches on SAB in patients with hematological diseases or other underlying diseases(4, 5, 26, 27). SAB is prone to causing severe deep-seated metastatic infections or deep-site infection recurrence (such as endocarditis and osteomyelitis), thus leading to readmissions or late mortality. Zeylemaker et al investigated the late infectious complications of catheter-related SAB (such as endocarditis, osteomyelitis, and metastatic abscesses) over a 1-year follow-up period in 49 adult patients(28). Twenty-four patients had developed a complication, among whom 11 (45.8%), 9 (37.5%), 1 (4.2%), and 6 (25%) patients developing complications at < 1 month, 1–3 months, 3–6 months, and > 6 months after their period of active disease, respectively(28) . This study supports the necessity of extended follow-up for patients with SAB, thus 90-day freedom from relapse or survival may be a more meaningful marker of cure for SAB. To account for potential confounders, we adjusted for key clinical variables, including underlying hematologic disease, severity of illness and treatment regimens, Charlson comorbidity index in our multivariable analysis. Additionally, to mitigate immortal time bias, we defined time zero as the date of initial positive blood culture rather than the completion of SAB treatment.

Minor comments:

Comment 1: *“Line 93: Why was an age cut-off of ≥ 14 years old utilized versus a standard adult population of ≥ 18 years old?”*

Response 1: Thank you for your question. In our study, we defined adults as patients aged ≥ 14 years based on our center’s clinical practice, where adolescents aged 14 and above are managed similarly to adults, especially in the context of infections like SAB. This classification aligns with our institutional guidelines and previous studies on bloodstream infections in hematological patients(29). Additionally, previous studies on bloodstream

infections in hematological patients have also included individuals aged 14 or 15 years and older as adults(30, 31). Given the immunocompromised nature of these patients, their disease course and treatment strategies are more comparable to adult patients rather than pediatric populations.

Comment 2: *“Line 126: Who completed the assessment regarding the adequacy of fluid resuscitation, and how was it assessed?”*

Response 2: Thank you again for your insightful question. The adequacy of fluid resuscitation was assessed by the treating physicians based on institutional protocols and international sepsis guidelines (32). Typically, patients received at least 30 mL/kg of crystalloid fluids, or more if clinically indicated, with continuous monitoring of hemodynamic parameters such as blood pressure response, lactate clearance, and urine output. If the mean arterial pressure (MAP) remained below 65 mmHg despite fluid resuscitation, vasopressors were initiated as per standard practice. Successful resuscitation was defined as reaching a CVP of 8–12 mmHg and maintaining a MAP of ≥ 65 mmHg. We have clarified this in the Methods section to enhance transparency (Line 150-153 in the Marked-up manuscript).

Comment 3: *“Line 127: Who assessed the infection source (treating team, or retrospectively by study team)?”*

Response 3: Thank you for your insightful question. The infection source was initially determined by the treating physicians during clinical management and recorded in the medical charts. For this study, the research team retrospectively reviewed the medical records and confirmed the infection source according to the Centers for Disease Control (CDC) definitions(33), including clinical signs, microbiological findings, and imaging results. In cases of uncertainty, adjudication was performed by two independent investigators. We have clarified this in the Methods section for transparency (Line 153-158 in the Marked-up manuscript).

Comment 4: *“Lines 142-145: References to support this statistical approach and execution should be included”*

Response 4: We appreciate the reviewer’s comment. The inverse probability of treatment weighting (IPTW) was conducted using stabilized weights derived from propensity scores to balance baseline characteristics between short-course and long-course antibiotic groups. We assessed covariate balance after weighting using standardized mean differences (SMDs), ensuring all were below the recommended threshold of 0.1. We will add appropriate references (e.g., Grool, A.M, 2016 (34); Ren, Q.W, 2021(35)) to support this statistical approach in the Methods section (Line 175 in the Marked-up manuscript).

Comment 5: *“Line 147: Why was GVHD (skin, GI, other sites) not included as a covariate?”*

Response 5: We appreciate the reviewer’s suggestion. In our study, the history of hematopoietic stem cell transplantation (HSCT) was defined as a history of transplantation within 1 year or the presence of acute or chronic graft versus host diseases (GVHD) during SAB, as both conditions are closely linked and represent the overall impact of HSCT on patient outcomes. Adjusting for GVHD as an independent covariate in IPTW could lead to over-adjustment and potentially obscure the true impact of HSCT. In the cohort of uncomplicated SAB, 8 patients presented with GVHD when the SAB occurred, and no significant difference was observed between the short- and long-course groups (2 of 93 (2.1%) vs. 6 of 109 (5.2%)). Therefore, we considered HSCT as a comprehensive covariate rather than adjusting for GVHD independently.

Comment 6: *“Line 149: Why were these complications chosen, when others identified by references studies were not (e.g., catheter removal in Bello-Chavolla, et al)?”*

Response 6: Thank you for your insightful comment. In contrast to the findings of

Bello-Chavolla et al.(36), where catheter-related infections accounted for 56.4% (254/450) of cases, our study observed a much lower incidence (5%, 12/242). All patients diagnosed with catheter-related SAB had their catheters removed. Given this low prevalence, including catheter-related infection as a covariate in inverse probability of treatment weighting (IPTW) would have had minimal impact on balancing groups and could introduce statistical instability. Instead, we prioritized complications (oral mucositis, perianal infection, and pneumonia) that were more common in our cohort and had a stronger potential influence on patient outcomes.

The selection of covariates for IPTW was based on their relevance and prevalence in our study population. Unlike Bello-Chavolla et al., where abdominal sources of infection were a significant predictor of mortality, our study included only one case of splenic abscess, which was successfully treated without recurrence or mortality. Given its extremely low incidence, including this variable in IPTW would not have been statistically meaningful. Additionally, blood glucose levels at the time of SAB onset were not systematically recorded in our dataset, preventing their inclusion in IPTW adjustment. However, a history of diabetes was incorporated as a covariate in the model, which serves as a surrogate marker for the potential impact of hyperglycemia on bloodstream infections.

Comment 7: *“Line 97 & 174: In line 97, patients with metastatic infections were noted as excluded from the definition of uncomplicated SAB. Please clarify how patients with metastatic infection are reported in the results.”*

Response 7: Thank you for your comment. In this study, metastatic infection was defined as definite infective endocarditis fulfilling the modified Duke criteria(37) or the development of secondary infection at a site distant from the initial focus of infection(38), which has been added to the Methods section (Line 159-161 in the Marked-up manuscript). Metastatic infection was diagnosed based on clinical signs, microbiological findings, and imaging results, consistent with standard definitions in SAB studies(38, 39). Specifically, patients were identified as having metastatic infections if additional infection sites were confirmed by imaging (e.g., CT, MRI) or microbiological evidence from normally sterile sites. These cases

were reported separately in the Results section (Lines 207-210 in the Marked-up manuscript).

Comment 8: *“Table 1: Was neutrophil recovery at the time of therapy cessation included in the study, and if not, please discuss why, given its potential association with treatment duration selection.”*

Response 8: Thank you for underlining this deficiency. In the original analysis, neutrophil recovery at the time of therapy cessation was not included. However, based on its potential association with treatment duration selection and clinical outcomes, we have now incorporated it as a study variable in the revised analysis. The updated results are reflected accordingly in Table 1 and Table 2 in the Marked-up manuscript.

1. Thorlacius-Ussing L, Sandholdt H, Nissen J, Rasmussen J, Skov R, Frimodt-Møller N, Dahl Knudsen J, Østergaard C, Benfield T. 2021. Comparable Outcomes of Short-Course and Prolonged-Course Therapy in Selected Cases of Methicillin-Susceptible *Staphylococcus aureus* Bacteremia: A Pooled Cohort Study. *Clinical Infectious Diseases* 73:866-872.
2. Kouijzer IJE, Fowler VG, Jr., Ten Oever J. 2023. Redefining *Staphylococcus aureus* bacteremia: A structured approach guiding diagnostic and therapeutic management. *J Infect* 86:9-13.
3. van der Vaart TW, Prins JM, Soetekouw R, van Twillert G, Veenstra J, Herpers BL, Rozemeijer W, Jansen RR, Bonten MJM, van der Meer JTM. 2022. Prediction Rules for Ruling Out Endocarditis in Patients With *Staphylococcus aureus* Bacteremia. *Clin Infect Dis* 74(8):1442-1449.

4. Ryu B-H, Lee SC, Kim M, Eom Y, Jung J, Kim MJ, Sung H, Kim M-N, Kim S-H, Lee S-O, Choi S-H, Woo JH, Kim YS, Chong YP. 2020. Impact of neutropenia on the clinical outcomes of *Staphylococcus aureus* bacteremia in patients with hematologic malignancies: a 10-year experience in a tertiary care hospital. *European Journal of Clinical Microbiology & Infectious Diseases* 39:937-943.
5. Camp J, Filla T, Glaubitz L, Kaasch AJ, Fuchs F, Scarborough M, Kim HB, Tilley R, Liao C-H, Edgeworth J, Nsutebu E, López-Cortés LE, Morata L, Llewelyn MJ, Fowler VG, Thwaites G, Seifert H, Kern WV, Rieg S. 2022. Impact of neutropenia on clinical manifestations and outcome of *Staphylococcus aureus* bloodstream infection: a propensity score-based overlap weight analysis in two large, prospectively evaluated cohorts. *Clinical Microbiology and Infection* 28:1149.e1-1149.e9.
6. Venditti M, Falcone M, Micozzi A, Carfagna P, Taglietti F, Serra PF, Martino P. 2003. *Staphylococcus aureus* bacteremia in patients with hematologic malignancies: a retrospective case-control study. *Haematologica* 88(8):923-30.
7. Curran J, Lo J, Leung V, Brown K, Schwartz KL, Daneman N, Garber G, Wu JHC, Langford BJ. 2022. Estimating daily antibiotic harms: an umbrella review with individual study meta-analysis. *Clin Microbiol Infect* 28:479-490.
8. Lopez-Cortes LE, Del Toro MD, Galvez-Acebal J, Bereciartua-Bastarrica E, Farinas MC, Sanz-Franco M, Natera C, Corzo JE, Lomas JM, Pasquau J, Del Arco A, Martinez MP, Romero A, Muniain MA, de Cueto M, Pascual A,

- Rodriguez-Bano J, group RS. 2013. Impact of an evidence-based bundle intervention in the quality-of-care management and outcome of *Staphylococcus aureus* bacteremia. *Clin Infect Dis* 57:1225-33.
9. Chong YP, Moon SM, Bang K-M, Park HJ, Park S-Y, Kim M-N, Park K-H, Kim S-H, Lee S-O, Choi S-H, Jeong J-Y, Woo JH, Kim YS. 2013. Treatment Duration for Uncomplicated *Staphylococcus aureus* Bacteremia To Prevent Relapse: Analysis of a Prospective Observational Cohort Study. *Antimicrobial Agents and Chemotherapy* 57:1150-1156.
 10. Wiggers JB, Xiong W, Daneman N. 2016. Sending repeat cultures: is there a role in the management of bacteremic episodes? (SCRIBE study). *BMC Infect Dis* 16:286.
 11. Tamma PD, Avdic E, Li DX, Dzintars K, Cosgrove SE. 2017. Association of Adverse Events With Antibiotic Use in Hospitalized Patients. *JAMA Internal Medicine* 177.
 12. Thwaites GE, Gant V. 2011. Are bloodstream leukocytes Trojan Horses for the metastasis of *Staphylococcus aureus*? *Nature Reviews Microbiology* 9:215-222.
 13. Zhu H, Jin H, Zhang C, Yuan T. 2020. Intestinal methicillin-resistant *Staphylococcus aureus* causes prosthetic infection via 'Trojan Horse' mechanism: Evidence from a rat model. *Bone & Joint Research* 9:152-161.
 14. Krezalek MA, Hyoju S, Zaborin A, Okafor E, Chandrasekar L, Bindokas V,

- Guyton K, Montgomery CP, Daum RS, Zaborina O, Boyle-Vavra S, Alverdy JC. 2018. Can Methicillin-resistant *Staphylococcus aureus* Silently Travel From the Gut to the Wound and Cause Postoperative Infection? Modeling the “Trojan Horse Hypothesis”. *Annals of Surgery* 267:749-758.
15. Lehar SM, Pillow T, Xu M, Staben L, Kajihara KK, Vandlen R, DePalatis L, Raab H, Hazenbos WL, Hiroshi Morisaki J, Kim J, Park S, Darwish M, Lee B-C, Hernandez H, Loyet KM, Lupardus P, Fong R, Yan D, Chalouni C, Luis E, Khalfin Y, Plise E, Cheong J, Lyssikatos JP, Strandh M, Koefoed K, Andersen PS, Flygare JA, Wah Tan M, Brown EJ, Mariathasan S. 2015. Novel antibody–antibiotic conjugate eliminates intracellular *S. aureus*. *Nature* 527:323-328.
16. van der Vaart TW, Prins JM, Goorhuis A, Lemkes BA, Sigaloff KCE, Spoorenberg V, Stijnis C, Bonten MJM, van der Meer JTM. 2024. The Utility of Risk Factors to Define Complicated *Staphylococcus aureus* Bacteremia in a Setting With Low Methicillin-Resistant *S. aureus* Prevalence. *Clinical Infectious Diseases* 78:846-854.
17. Hendriks MMC, Schwersen KSA, Kleij A, Berrevoets MAH, de Jong E, van Wijngaarden P, Ammerlaan HSM, Vos A, van Assen S, Slieker K, Gisolf JH, Netea MG, Ten Oever J, Kouijzer IJE. 2024. Low-Risk *Staphylococcus aureus* Bacteremia Patients Do Not Require Routine Diagnostic Imaging: A Multicenter, Retrospective, Cohort Study. *Clin Infect Dis* 79:43-51.
18. Barton T, Moir S, Rehmani H, Woolley I, Korman TM, Stuart RL. 2016. Low rates of endocarditis in healthcare-associated *Staphylococcus aureus*

bacteremia suggest that echocardiography might not always be required. Eur J Clin Microbiol Infect Dis 35:49-55.

19. Heriot GS, Cronin K, Tong SYC, Cheng AC, Liew D. 2017. Criteria for Identifying Patients With Staphylococcus aureus Bacteremia Who Are at Low Risk of Endocarditis: A Systematic Review. Open Forum Infect Dis 4:ofx261.
20. Arman G, Zeyad M, Qindah B, Abu Taha A, Amer R, Abutaha S, Koni AA, Zyoud SH. 2022. Frequency of microbial isolates and pattern of antimicrobial resistance in patients with hematological malignancies: a cross-sectional study from Palestine. BMC Infect Dis 22:146.
21. Kara Ö, Zarakolu P, Aşcıoğlu S, Etgül S, Uz B, Büyükaşık Y, Akova M. 2015. Epidemiology and emerging resistance in bacterial bloodstream infections in patients with hematologic malignancies. Infectious Diseases 47:686-693.
22. Peri AM, Edwards F, Henden A, Harris PNA, Chatfield MD, Paterson DL, Laupland KB. 2023. Bloodstream infections in neutropenic and non-neutropenic patients with haematological malignancies: epidemiological trends and clinical outcomes in Queensland, Australia over the last 20 years. Clin Exp Med 23:4563-4573.
23. Zhou S, Xin C, Liu W. 2025. Sequential Therapy of Linezolid and Contezolid to Treat Hematogenous Lung Abscess Caused by Staphylococcus aureus in a Congenital Cerebral Hypoplasia Patient: A Case Report. Infect Drug Resist 18:253-260.
24. Yang W, Li X, Chen J, Zhang G, Li J, Zhang J, Wang T, Kang W, Gao H,

- Zhang Z, Liu Y, Xiao Y, Xie Y, Zhao J, Mao L, Sun Z, Li G, Jia W, Song G, Shan B, Yu Y, Sun G, Xu Y, Liu Y. 2024. Multicentre evaluation of in vitro activity of contezolid against drug-resistant *Staphylococcus* and *Enterococcus*. *J Antimicrob Chemother* 79:3132-3141.
25. Chen Y, Ren J, Li F, Ye X, Wu Y. 2024. The antibiotic therapy containing contezolid successfully treated methicillin-sensitive *Staphylococcus aureus* infective endocarditis accompanied with cerebrovascular complications. *BMC Infect Dis* 24:1301.
26. Perez-Rodriguez MT, Sousa A, Moreno-Flores A, Longueira R, Dieguez P, Suarez M, Lima O, Vasallo FJ, Alvarez-Fernandez M, Crespo M. 2021. The benefits and safety of oral sequential antibiotic therapy in non-complicated and complicated *Staphylococcus aureus* bacteremia. *Int J Infect Dis* 102:554-560.
27. Nambiar K, Seifert H, Rieg S, Kern WV, Scarborough M, Gordon NC, Kim HB, Song KH, Tilley R, Gott H, Liao CH, Edgeworth J, Nsutebu E, Lopez-Cortes LE, Morata L, Walker AS, Thwaites G, Llewelyn MJ, Kaasch AJ, International *Staphylococcus aureus* collaboration study g, the ESGfBI, Sepsis. 2018. Survival following *Staphylococcus aureus* bloodstream infection: A prospective multinational cohort study assessing the impact of place of care. *J Infect* 77:516-525.
28. Zeylemaker MM, Jaspers CA, van Kraaij MG, Visser MR, Hoepelman IM. 2001. Long-term infectious complications and their relation to treatment duration in catheter-related *Staphylococcus aureus* bacteremia. *Eur J Clin*

Microbiol Infect Dis 20:380-4.

29. Guo W, Lin Q, Li J, Feng X, Zhen S, Mi Y, Zheng Y, Zhang F, Xiao Z, Jiang E, Han M, Wang J, Feng S. 2025. *Stenotrophomonas maltophilia* bacteremia in adult patients with hematological diseases: clinical characteristics and risk factors for 28-day mortality. *Microbiol Spectr* 13:e0101124.
30. Gedik H, Yildirmak T, Simsek F, Kanturk A, Arica D, Aydin D, Yokus O, Demirel N, Arabaci C. 2014. Vancomycin-resistant enterococci colonization and bacteremia in patients with hematological malignancies. *J Infect Dev Ctries* 8:1113-8.
31. Kang CI, Song JH, Chung DR, Peck KR, Yeom JS, Son JS, Wi YM, Korean Network for Study on Infectious D. 2012. Bloodstream infections in adult patients with cancer: clinical features and pathogenic significance of *Staphylococcus aureus* bacteremia. *Support Care Cancer* 20:2371-8.
32. Rhodes A, Evans LE, Alhazzani W, Levy MM, Antonelli M, Ferrer R, Kumar A, Sevransky JE, Sprung CL, Nunnally ME, Rochweg B, Rubenfeld GD, Angus DC, Annane D, Beale RJ, Bellingham GJ, Bernard GR, Chiche J-D, Coopersmith C, De Backer DP, French CJ, Fujishima S, Gerlach H, Hidalgo JL, Hollenberg SM, Jones AE, Karnad DR, Kleinpell RM, Koh Y, Lisboa TC, Machado FR, Marini JJ, Marshall JC, Mazuski JE, McIntyre LA, McLean AS, Mehta S, Moreno RP, Myburgh J, Navalesi P, Nishida O, Osborn TM, Perner A, Plunkett CM, Ranieri M, Schorr CA, Seckel MA, Seymour CW, Shieh L, Shukri KA, et al. 2017. Surviving Sepsis Campaign: International Guidelines

for Management of Sepsis and Septic Shock: 2016. *Intensive Care Medicine* 43:304-377.

33. Garner JS, Jarvis WR, Emori TG, Horan TC, Hughes JM. 1988. CDC definitions for nosocomial infections, 1988. *Am J Infect Control* 16:128-40.
34. Grool AM, Aglipay M, Momoli F, Meehan WP, Freedman SB, Yeates KO, Gravel J, Gagnon I, Boutis K, Meeuwisse W, Barrowman N, Ledoux A-A, Osmond MH, Zemek R. 2016. Association Between Early Participation in Physical Activity Following Acute Concussion and Persistent Postconcussive Symptoms in Children and Adolescents. *Jama* 316.
35. Ren QW, Yu SY, Teng TK, Li X, Cheung KS, Wu MZ, Li HL, Wong PF, Tse HF, Lam CSP, Yiu KH. 2021. Statin associated lower cancer risk and related mortality in patients with heart failure. *Eur Heart J* 42:3049-3059.
36. Bello-Chavolla OY, Bahena-Lopez JP, Garciadiego-Fosass P, Volkow P, Garcia-Horton A, Velazquez-Acosta C, Vilar-Compte D. 2018. Bloodstream infection caused by *S. aureus* in patients with cancer: a 10-year longitudinal single-center study. *Supportive Care in Cancer* 26:4057-4065.
37. Li JS, Sexton DJ, Mick N, Nettles R, Fowler VG, Jr., Ryan T, Bashore T, Corey GR. 2000. Proposed modifications to the Duke criteria for the diagnosis of infective endocarditis. *Clin Infect Dis* 30:633-8.
38. Kim T, Lee SR, Park SY, Moon SM, Jung J, Kim MJ, Sung H, Kim MN, Kim SH, Choi SH, Lee SO, Kim YS, Song EH, Chong YP. 2024. Validation of a new risk stratification system-based management for methicillin-resistant

Staphylococcus aureus bacteraemia: analysis of a multicentre prospective study. *Eur J Clin Microbiol Infect Dis* 43:841-851.

39. Fowler VG, Jr., Olsen MK, Corey GR, Woods CW, Cabell CH, Reller LB, Cheng AC, Dudley T, Oddone EZ. 2003. Clinical identifiers of complicated Staphylococcus aureus bacteremia. *Arch Intern Med* 163:2066-72.

Re: Spectrum02325-24R1 (**Clinical Characteristics and Efficacy of Short-Course Antibiotic Therapy for *Staphylococcus aureus* Bacteremia in Hematological Patients**)

Dear Dr. Sizhou Feng:

Thank you for the privilege of reviewing your work. Below you will find my comments, instructions from the Spectrum editorial office, and the reviewer comments.

Revision Guidelines

Sincerely,
Bonnie Prokesch
Editor
Microbiology Spectrum

Reviewer #2 (Comments for the Author):

The authors provided thoughtful responses to the comments and questions with the original submission.

Reviewer #3 (Comments for the Author):

Methods - Line 187 (marked up manuscript) refers to composite outcomes, however, from the text, it is unclear which outcome(s) you are referring to. Please clarify in the manuscript.

Results - Please add information regarding the consultation of infectious diseases physicians for all patients and the rate of echocardiography to the results section.

Discussion

- Lines 242-251 (marked up manuscript), please clarify if the studies you discuss were focused on hematologic malignancy patients, and if not, describe the proportion of the study population that was (if available)
- Lines 347-352, please clarify if you are referring to the results of your present study
- Line 349, please include the number of deep-seated infections

Limitations - Please acknowledge the lack of systematic TDM and that infection-related mortality was not assessed as limitations in the manuscript text.

Supplementary Table 1 - Why is imipenem reported for MRSA treatment when this antimicrobial does not have activity?

Re: Spectrum02325-24 “Clinical Characteristics and Efficacy of Short-Course Antibiotic Therapy for Staphylococcus aureus Bacteremia in Hematological Patients”

Dear editors,

Thank you once again for your continued evaluation of our manuscript and for the detailed and constructive comments provided by you and the reviewers. We greatly appreciate the time and effort devoted to helping us improve our work.

We have carefully addressed all the concerns raised and revised the manuscript accordingly. Reviewer comments are presented in italicized font and numbered for clarity, with our point-by-point responses provided in regular font. All modifications or additions have been incorporated into the revised manuscript. The revisions have been uploaded in the Marked-up Manuscript with changes highlighted in yellow.

Supplementary tables have also been redrawn and re-uploaded as requested.

We sincerely hope that the revised manuscript satisfactorily addresses all the concerns and meets the standards for publication. We look forward to your feedback and the opportunity to publish our work in your esteemed journal.

Sincerely,

The Authors

Email: doctor_szhfeng@163.com; yangnuobing@ihcams.ac.cn

Reply to Reviewer 3:

Comment 1: *“Methods - Line 187 (marked up manuscript) refers to composite outcomes, however, from the text, it is unclear which outcome(s) you are referring to. Please clarify in the manuscript.”*

Response 1: Thank you for pointing this out. The mention of "composite outcomes" was an oversight and does not accurately reflect the outcomes analyzed in our study. We have removed the word "composite" from the text in Line 187 of the marked-up manuscript to avoid confusion.

Comment 2: *“Results - Please add information regarding the consultation of infectious diseases physicians for all patients and the rate of echocardiography to the results section.”*

Response 2: Thank you very much for your suggestions. We have added information regarding infectious diseases physician consultations and the rate of echocardiography to the Results section. Specifically, we added the following sentence: “All patients received bedside consultations from infectious diseases physicians. And 37 (15.3%) patients underwent echocardiography.” This content has been incorporated into **Lines 201–202** and highlighted in yellow in the Marked-up Manuscript for your review.

Comment 3: *“Discussion - Lines 242-251 (marked up manuscript), please clarify if the studies you discuss were focused on hematologic malignancy patients, and if not,*

describe the proportion of the study population that was (if available)."

Response 3: Thank you again for your insightful suggestion. The two studies discussed in this section were conducted in general patient populations with various underlying conditions and were not specifically focused on patients with hematologic malignancies. While both studies reported the proportion of patients with cancers (11.3% (n=33) and 17.2% (n=80), respectively), they did not provide specific data on the proportion of patients with hematologic malignancies. We have clarified this point in the Discussion section. The following content has been added to **Lines 277–290** and highlighted in yellow in the Marked-up Manuscript for your reference:

"In a study of 293 patients from the general population with SAB, 45 (15.4%) developed metastatic infections and 10 (3.4%) had endocarditis(1). Among them, 82 (28.0%) died during hospitalization, with 68 (83%) of these deaths occurring within the first 30 days after the onset of SAB(1). Russell et al also reported findings from 464 SAB patients with various underlying conditions, not limited to hematologic diseases(2). They identified 134 (28.9%) participants with apparent metastatic foci, with vertebral osteomyelitis (n = 54, 11.6%) being the most common, followed by endocarditis (n = 35, 7.5%) and septic arthritis (n = 29, 6.3%)(2). In their study, the recurrence rate and 90-day all-cause mortality rate were 1.9% (n = 9) and 28.0% (n = 130), respectively(2)."

These studies were included to provide a comparison between outcomes in the general SAB population and our cohort of hematologic patients. The incidence of

deep-seated metastatic infections (e.g., endocarditis, osteomyelitis) and the rates of mortality and recurrence were lower than or comparable to those reported in the general population. This suggests that the clinical outcomes of SAB in patients with hematologic diseases are not necessarily inferior to those in non-hematologic populations.

Comment 4: *“Discussion - Lines 347-352, please clarify if you are referring to the results of your present study.”*

Response 4: Thank you again for pointing this out. We agree that the interpretation in this part of the Discussion may have caused confusion regarding whether it referred to our own data or external evidence. To avoid potential misunderstanding, we have removed this section from the revised manuscript.

The deleted content is as follows:

“In cases with a clearly identified source of infection, the majority involved curable or eradicable foci (mainly skin or soft tissue), while deep-seated infectious foci were really rare. A small proportion of patients did not completely recover from neutropenia when discontinuing antibiotics. They were evenly distributed between the long-course and short-course groups but seemed more prone to experience adverse outcomes, suggesting that clinicians should consider approaches beyond simply extending the antibiotic duration.”

In the Revised Manuscript, this part has been removed. In the Marked-up Manuscript,

this deleted section has been highlighted in green for your reference (Line 357-362).

Comment 5: *“Discussion - Line 349, please include the number of deep-seated infections.”*

Response 5: Thank you for highlighting this important issue. We have revised the manuscript to clarify the distribution of infection sources in the Result section (Line 205-209 in the Marked-up Manuscript). Specifically, we now state that the leading sources were primary (n=141, 58.3%) and soft tissue or skin infections (n=63, 26.0%), followed by respiratory tract-associated (n=15, 6.2%), PICC-associated (n=12, 5.0%), and gastrointestinal tract-associated infections (n=9, 3.7%). In addition, we have included one case of a splenic abscess identified as the source of *Staphylococcus aureus* bacteremia, for which the patient underwent splenectomy (Line 207-209 in the Marked-up manuscript).

Comment 6: *“Limitations - Please acknowledge the lack of systematic TDM and that infection-related mortality was not assessed as limitations in the manuscript text.”*

Response 6: Thank you for your suggestions. We have acknowledged both the lack of systematic TDM and the absence of infection-related mortality assessment as limitations.

The following sentence has been added to the Limitations section:

“In addition, therapeutic drug monitoring was not systematically implemented during the study period (2013–2023), which may have affected the evaluation of antibiotic efficacy and toxicity. Finally, infection-related mortality was not specifically assessed, and our outcome analysis was limited to all-cause mortality.”

These additions are highlighted in the Marked-up Manuscript (Line 377-380) for your reference.

Comment 7: *“Supplementary Table 1 - Why is imipenem reported for MRSA treatment when this antimicrobial does not have activity?”*

Response 7: Thank you very much for pointing this out. Imipenem was initially listed in Supplementary Table 1 because it was administered in some patients who concurrently had Gram-negative bacteremia, suspected Gram-negative pneumonia, or other concomitant infections caused by Gram-negative organisms. However, it was not used as targeted therapy for MRSA. To avoid confusion, we have removed imipenem from Supplementary Table 1 in the revised manuscript.

Gram-negative bacteria have recently been reported to be a more frequent cause of bacteremia than Gram-positive bacteria in hematological patients, with Enterobacterales being the most commonly isolated pathogens, followed by *Klebsiella pneumoniae* and *Pseudomonas aeruginosa*(3-6). Moreover, infections caused by Gram-negative bacteria are usually associated with higher morbidity and mortality(7).

In a study by Guarana et al. involving 1305 episodes of febrile neutropenia in patients with hematologic malignancies, septic shock occurred in 42 episodes (3.2%) and early death in 15 cases (1.1%)(8). Bacteremia due to *Escherichia coli* (odds ratio [OR], 8.47; 95% confidence interval [CI], 4.08-17.55; $P < 0.001$), *Enterobacter* sp. (OR, 7.53; 95% CI, 1.60–35.33; $P = 0.01$), and *Acinetobacter* sp. (OR, 6.95; 95% CI, 1.49-32.36; $P = 0.01$) were significant predictors of shock, while *Klebsiella pneumoniae* bacteremia was associated with early death (OR, 5.91; 95% CI, 1.11–31.47; $P = 0.03$)(8). Similarly, Jung et al. identified 109 bacteria isolates in 133 patients with neutropenic septic shock, with Gram-negative organisms accounting for 77.1% ($n = 84$), highlighting their predominance in this setting(9). In another study, Treçarichi et al. reported 668 bacterial isolates from 575 bloodstream infection episodes in adults with hematologic malignancies(10). The 21-day mortality rate was significantly higher in infections caused by Gram-negative bacteria compared to those caused by Gram-positive organisms (47/278, 16.9% vs. 12/212, 5.6%; $P < 0.001$), with particularly high mortality observed in cases of bacteremia due to *K. pneumoniae*, *P. aeruginosa*, and *Acinetobacter baumannii*(10).

Appropriate empirical treatment can significantly improve the prognosis(8). Guideline recommended that high-risk febrile neutropenic patients receive intravenous empirical antibiotic therapy with an anti-pseudomonal β -lactam agent, such as cefepime, a carbapenem (meropenem or imipenem-cilastatin), or piperacillin-tazobactam(11). Carbapenems are active against methicillin-susceptible *Staphylococcus aureus* (MSSA)(12), and in vitro studies have also demonstrated their synergistic activity with

vancomycin against methicillin-resistant *Staphylococcus aureus* (MRSA)(13). In our cohort, when MRSA bacteremia was identified, some patients had concurrent pulmonary infections or bloodstream infections caused by Gram-negative bacteria; therefore, imipenem were not discontinued despite their lack of activity against MRSA.

1. Mylotte JM, Tayara A. 2000. Staphylococcus aureus Bacteremia: Predictors of 30-Day Mortality in a Large Cohort. *Clinical Infectious Diseases* 31:1170-1174.
2. Russell CD, Berry K, Cooper G, Sim W, Lee RS, Gan TY, Donlon W, Besu A, Heppenstall E, Tysall L, Robb A, Dewar S, Smith A, Fowler VG. 2024. Distinct Clinical Endpoints of Staphylococcus aureus Bacteraemia Complicate Assessment of Outcome. *Clinical Infectious Diseases* doi:10.1093/cid/ciae281.
3. Trecarichi EM, Giuliano G, Cattaneo C, Ballanti S, Criscuolo M, Candoni A, Marchesi F, Laurino M, Dargenio M, Fanci R, Cefalo M, Delia M, Spolzino A, Maracci L, Bonuomo V, Busca A, Principe MID, Daffini R, Simonetti E, Dragonetti G, Zannier ME, Pagano L, Tumbarello M, Hematologic Malignancies Associated Bloodstream Infections Surveillance registry - Sorveglianza Epidemiologica Infezioni Fungine in Emopatie Maligne group I. 2023. Bloodstream infections due to Gram-negative bacteria in patients with hematologic malignancies: updated epidemiology and risk factors for

multidrug-resistant strains in an Italian perspective survey. *Int J Antimicrob Agents* 61:106806.

4. Averbuch D, Tridello G, Hoek J, Mikulska M, Akan H, Yanez San Segundo L, Pabst T, Ozcelik T, Klyasova G, Donnini I, Wu D, Gulbas Z, Zuckerman T, Botelho de Sousa A, Beguin Y, Xhaard A, Bachy E, Ljungman P, de la Camara R, Rascon J, Ruiz Camps I, Vitek A, Patriarca F, Cudillo L, Vrhovac R, Shaw PJ, Wolfs T, O'Brien T, Avni B, Silling G, Al Sabty F, Graphakos S, Sankelo M, Sengeloev H, Pillai S, Matthes S, Melanthiou F, Iacobelli S, Styczynski J, Engelhard D, Cesaro S. 2017. Antimicrobial Resistance in Gram-Negative Rods Causing Bacteremia in Hematopoietic Stem Cell Transplant Recipients: Intercontinental Prospective Study of the Infectious Diseases Working Party of the European Bone Marrow Transplantation Group. *Clin Infect Dis* 65:1819-1828.
5. Cruz-Vargas SA, Garcia-Munoz L, Cuervo-Maldonado SI, Alvarez-Moreno CA, Saavedra-Trujillo CH, Alvarez-Rodriguez JC, Arango-Gutierrez A, Gomez-Rincon JC, Garcia-Guzman K, Leal AL, Garzon-Herazo J, Martinez-Vernaza S, Guevara FO, Jimenez-Cetina LP, Mora LM, Saavedra SY, Cortes JA. 2023. Molecular and Clinical Data of Antimicrobial Resistance in Microorganisms Producing Bacteremia in a Multicentric Cohort of Patients with Cancer in a Latin American Country. *Microorganisms* 11.
6. El Omri H, Padmanabhan R, Taha RY, Kassem N, Elsabah H, Ellahie AY, Santimano AJJ, Al-Maslmani MA, Omrani AS, Elomri A, El Omri A. 2024.

- Dissecting bloodstream infections in febrile neutropenic patients with hematological malignancies, a decade-long single center retrospective observational study (2009-2019). *J Infect Public Health* 17:152-162.
7. Menzo SL, la Martire G, Ceccarelli G, Venditti M. 2015. New Insight on Epidemiology and Management of Bacterial Bloodstream Infection in Patients with Hematological Malignancies. *Mediterr J Hematol Infect Dis* 7:e2015044.
 8. Guarana M, Nucci M, Nouér SA. 2019. Shock and Early Death in Hematologic Patients with Febrile Neutropenia. *Antimicrobial Agents and Chemotherapy* 63.
 9. Jung SM, Kim YJ, Ryoo SM, Sohn CH, Seo DW, Lim KS, Kim WY. 2020. Cancer patients with neutropenic septic shock: etiology and antimicrobial resistance. *Korean J Intern Med* 35:979-987.
 10. Treccarichi EM, Pagano L, Candoni A, Pastore D, Cattaneo C, Fanci R, Nosari A, Caira M, Spadea A, Busca A, Vianelli N, Tumbarello M, HeMabis Registry-Seifem Group I. 2015. Current epidemiology and antimicrobial resistance data for bacterial bloodstream infections in patients with hematologic malignancies: an Italian multicentre prospective survey. *Clin Microbiol Infect* 21:337-43.
 11. Freifeld AG, Bow EJ, Sepkowitz KA, Boeckh MJ, Ito JI, Mullen CA, Raad II, Rolston KV, Young J-AH, Wingard JR. 2011. Clinical Practice Guideline for the Use of Antimicrobial Agents in Neutropenic Patients with Cancer: 2010 Update by the Infectious Diseases Society of America. *Clinical Infectious*

Diseases 52:e56-e93.

12. Hawkey PM, Livermore DM. 2012. Carbapenem antibiotics for serious infections. *BMJ* 344:e3236.
13. Jankeel A, Perez-Parra G, Khetarpal AK, Alvarado IA, Nizet V, Sakoulas G, Ulloa ER. 2025. Enhanced Killing of Methicillin-Resistant *Staphylococcus aureus* with Ceftaroline or Vancomycin in Combination with Carbapenems. *J Infect Dis* doi:10.1093/infdis/jiaf010.

Re: Spectrum02325-24R2 (**Clinical Characteristics and Efficacy of Short-Course Antibiotic Therapy for Staphylococcus aureus Bacteremia in Hematological Patients**)

Dear Dr. Sizhou Feng:

Your manuscript has been accepted, and I am forwarding it to the ASM production staff for publication. Your paper will first be checked to make sure all elements meet the technical requirements. ASM staff will contact you if anything needs to be revised before copyediting and production can begin. Otherwise, you will be notified when your proofs are ready to be viewed.

Sincerely,
Bonnie Prokesch
Editor
Microbiology Spectrum